# Tau forms synaptic nano-biomolecular condensates controlling the dynamic clustering of recycling synaptic vesicles

Shanley F. Longfield[1], Mahdie Mollazade[1], Tristan P. Wallis [1],
Rachel S. Gormal [1], Merja Joensuu [1,5], Jesse R. Wark [2],
Ashley J. van Waardenberg[3], Christopher Small[1], Mark E. Graham [2],
Frédéric A. Meunier [1,4] ✉ & Ramón Martínez-Mármol [1] ✉

Neuronal communication relies on the release of neurotransmitters from various populations of synaptic vesicles. Despite displaying vastly different release probabilities and mobilities, the reserve and recycling pool of vesicles co-exist within a single cluster suggesting that small synaptic biomolecular condensates could regulate their nanoscale distribution. Here, we performed a large-scale activity-dependent phosphoproteome analysis of hippocampal neurons in vitro and identified Tau as a highly phosphorylated and disordered candidate protein. Single-molecule super-resolution microscopy revealed that Tau undergoes liquid-liquid phase separation to generate presynaptic nanoclusters whose density and number are regulated by activity. This activity-dependent diffusion process allows Tau to translocate into the presynapse where it forms biomolecular condensates, to selectively control the mobility of recycling vesicles. Tau, therefore, forms presynaptic nano-biomolecular condensates that regulate the nanoscale organization of synaptic vesicles in an activity-dependent manner.

Neuronal communication hinges on the release of neurotransmitters, which are stored in synaptic vesicles (SVs) located at the presynapse. SVs are one of the smallest cellular organelles and, as such, are highly subjected to Brownian thermal energy, which tends to homogenize their distribution within the entire neuronal volume. Despite these significant randomization forces, SVs are not homogenously distributed and are instead highly enriched in nerve terminals, where they organize to form clusters that face their release sites at the active zone[1]. Electrophysiological analyses have demonstrated that SVs can be subdivided into distinct functional pools based on their ability to release neurotransmitters[2]. Around 10 to 20% of the total SV pool consists of recycling SVs. These vesicles rapidly reform from the plasma membrane following exocytosis and subsequent compensatory endocytosis[3]. The remaining 80 to 90% of vesicles consist of the reserve pool of SVs[1,4], which are not readily fusogenic, but can be recruited for fusion in response to strong and/or sustained stimulation[5]. Although the reserve and the recycling pools reside within the same vesicle cluster[6], early work suggested that they exhibit dramatically different mobilities and may adopt a multi-layer organization[7]. How these recycling SVs exhibit higher mobilities than reserve SVs while remaining in the same clusters is still unknown.

To answer these questions, direct imaging of both the recycling and the reserve pools is necessary. Presynapses are crowded and compacted regions, with a high concentration of SVs and associated

---

[1]Clem Jones Centre for Ageing Dementia Research (CJCADR), Queensland Brain Institute (QBI), The University of Queensland; St Lucia Campus, Brisbane, QLD 4072, Australia. [2]Synapse Proteomics, Children's Medical Research Institute (CMRI), The University of Sydney, 214 Hawkesbury Road, Westmead, NSW 2145, Australia. [3]i-Synapse, Cairns, QLD 4870, Australia. [4]School of Biomedical Science, The University of Queensland; St Lucia Campus, Brisbane, QLD 4072, Australia. [5]Present address: Australian Institute for Bioengineering and Nanotechnology, The University of Queensland; St Lucia Campus, Brisbane, QLD 4072, Australia. ✉e-mail: f.meunier@uq.edu.au; r.martinezmarmol@uq.edu.au

proteins, making their direct visualization extremely challenging. Moreover, the small size of SVs, far below the diffraction of light, has limited the use of classical optical microscopy techniques. Electron microscopy studies have demonstrated that the reserve and the recycling pools are dynamically intermixed[8]. Still, they could not provide real-time information on the dynamic organization of these essential SV pools. Single-molecule super-resolution microscopy has started to overcome these limitations, allowing the study of the dynamic distribution of SV pools at the nanoscale level[9,10]. SV cluster organization suggests the existence of anchoring mechanisms that prevent them from spreading out of the presynapse and throughout the axon. Synaptic phosphorylation was previously shown to dissipate SV clusters[11,12]. Recently, liquid-liquid phase separation (LLPS) has been hypothesized to control SV cluster organization by generating biomolecular condensates[13]. LLPS represents a unique mechanism of intracellular compartmentalization through membraneless structures[14,15], which typically occurs when multidomain proteins associate into biomolecular condensates by multivalent, low-affinity non-specific interactions[16] between intrinsically disordered regions (IDRs)[17].

Here, we reasoned that the mechanism controlling the nanoscale organization of SVs involves both an activity-dependent change in phosphorylation status and the ability to form biomolecular condensates via unstructured domains. We used mass spectrometry to identify highly disordered synaptic proteins whose phosphorylation status changes in response to stimulation in hippocampal synapses. Tau (MAPT) was identified as a highly-phosphorylated and disordered candidate protein, that has recently been shown to form biomolecular condensates[18]. Tau is a Type-II microtubule-associated protein that is predominantly expressed in the nervous system[19]. It functions as a versatile scaffolding protein[20] with established roles in neuronal plasticity[21], neurogenesis[22], neuronal migration[23] and most notably neurodegenerative diseases[24]. Tau is an unfolded protein that is highly soluble, but under certain circumstances, Tau can aggregate into insoluble toxic aggregates[25]. It undergoes multiple types of post-translational modifications under physiological conditions, but the most frequent of these is phosphorylation[26]. Tau can be phosphorylated at multiple sites[27], regulating its association with microtubules[28] and disrupting their organization[29]. Recently, Tau was shown to undergo LLPS, and pathological mutations of Tau affecting the formation of phase-separated condensates were also shown to influence synaptic transmission[30–32].

We combined various super-resolution approaches to show the specific nanoscale organization of different SV pools, identifying protein phosphorylation as an important mechanism that controls SV mobility. Analysis of SV mobility revealed the importance of Tau in controlling their organization. We then characterized the mobility of single Tau molecules in live neurons, revealing the profound effect that phase separation has on Tau's nanoscale organization. We found that Tau molecules are highly diffusible and have the ability to form dynamic nanoclusters. Importantly, Tau switches localization from the axon to nearby presynapses in an activity-dependent manner. Finally, we demonstrated that Tau selectively controls the mobility of recycling SVs and acts via the formation of transient presynaptic nano-biomolecular condensates. Our study provides new insights into the dynamic nanoscale organization of SVs in live hippocampal presynapses and reveals a specific role for Tau in forming nano-biomolecular condensates that control the mobility of the recycling pool of SVs.

## Results

We recently designed a super-resolution imaging technique named subdiffractional tracking of internalized molecules (sdTIM), which enabled tracking of individual recycling SVs in live hippocampal presynapses using single-molecule imaging[10]. SdTIM relies on very low concentrations of fluorescently labelled anti-GFP nanobodies (At647N-

GBP) being internalized into recycling SVs from hippocampal neurons expressing VAMP2-pHluorin (Fig. 1a, Supplementary Fig. S1a and Supplementary Video S1). To image the entirety of the SV pool, we used the vesicular glutamate transporter 1 (vGLUT1), an archetypical vesicular protein previously used to image SVs in central synapses[33–35]. vGLUT1 was fused to the photoconvertible fluorophore mEos2 (vGLUT1-mEos2) and was expressed in hippocampal neurons to perform single-particle tracking photoactivated localization microscopy (sptPALM) to image individual SVs from the total SV pool (Fig. 1b, Supplementary Fig. S1b and Supplementary Video S2). Analysis of SV mobility using these two strategies revealed completely different behaviors, with the recycling pool being much more mobile compared to the total pool, as measured by the average mean square displacement (MSD) (Fig. 1c and Supplementary Fig. S1c, d), the corresponding area under the curve MSD curve (AUC) (Fig. 1d), the diffusion coefficient distribution (Fig. 1e) and mobile fraction (Fig. 1f). Because it is estimated that 80 to 90% of the total pool of vesicles belongs to the reserve pool[1,2,36], the total pool is, therefore an adequate approximation for the reserve pool. To further investigate the heterogenous mobilities of these pools, we conducted variational Bayes single-particle tracking (vbSPT) analysis[37,38]. VbSPT analysis infers the number of hidden diffusive states and transitions between these states from the experimental data based on Bayesian model selection applied to hidden Markov modelling (HMM)[10,37]. Through HMM-Bayes analysis, thousands of trajectories from over a hundred synapses were each assigned to a discrete number of motion states[39,40], which were used to infer SV mobility states and state transitions. Our analysis indicated that individual SVs from both the recycling and the total pool can transition between three possible distinct diffusive states, which are categorized based on their apparent diffusion coefficients as immobile (State 1), confined (State 2) and highly mobile (State 3) (Fig. 1g, h). For both pools, the immobile state (State 1) is similar in terms of diffusion coefficient ($0.03–0.04\,\mu m^2/s$), state occupancy (~20%) and higher probability of transitioning to State 2. Importantly, the two pools differed in their confined and highly mobile states (Fig. 1g, h). In particular, most (~45%) of the recycling pool of SVs was mobile, with 35% being confined. The total pool, on the other hand, had more vesicles (>45%) in a confined state. Notably, the transition between immobile and mobile states occurs via the confined state (Fig. 1g, h). A representative example trajectory obtained from a recycling SV that oscillates between various mobility states is highlighted (Fig. 1i).

Clustering of SVs has been hypothesized to depend on the formation of condensates through interactions between protein IDRs[41–43]. Further, a multitude of synaptic proteins, particularly active zone scaffolding proteins, undergo a high level of activity-dependent phospho-regulation[44]. Therefore, both activity-dependent phosphorylation and the level of structural disorder represent key metrics worth joint consideration when identifying candidate proteins capable of controlling the mobility of recycling vesicles via LLPS. Accordingly, we used mass spectrometry analysis to obtain the phosphoproteome of mouse hippocampal neurons stimulated with high $K^+$, identifying 20,395 phosphopeptides, of which 4626 differed significantly in their phosphorylation status (threshold adjusted $P < 0.05$) in the stimulated condition compared to the resting control condition (Supplementary Data 1). An enrichment dataset of the proteins containing 500 of the most significant differentially phosphorylated peptides from stimulated hippocampal neurons was compared with the SynGO synaptic protein database[45] (Fig. 2a and Supplementary Fig. S2a). SynGO analysis revealed the presence of pre- and postsynaptic proteins, including postsynaptic density constituents and the presynaptic active zone, among the phosphorylated proteins identified (Supplementary Fig. S2a). It also highlighted a strong representation of proteins associated with mechanisms involved in the structural organization of the synapse and the recycling of SVs (Fig. 2a), supporting an activity-dependent phospho-regulation of proteins involved in

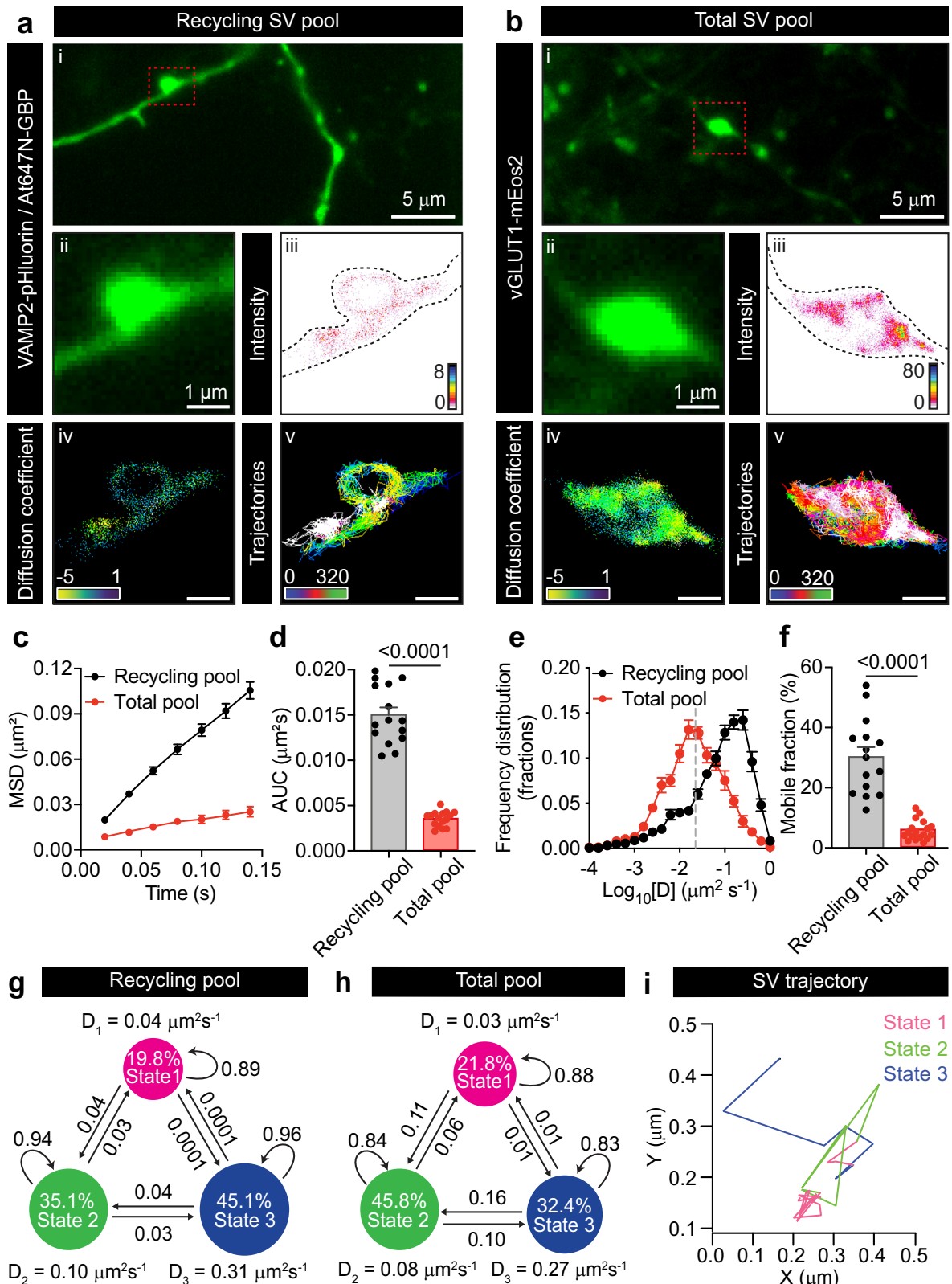

synapse organization. To further confirm a synaptic enrichment in our phosphoproteomic analysis, a total 4626 phosphopeptides from the obtained mouse hippocampal phosphoproteome were compared with a phosphoproteome dataset containing 1917 phosphopeptides from stimulated rat synaptosomes[44]. The sequence of each homologous protein from mouse neurons and rat synaptosomes was fully aligned to determine phosphosite detection and conversion. This analysis

indicated that synaptosome phosphosites responding to stimulation were also identified in the stimulated mouse total hippocampal neurons (Fig. 2b). The smaller number of phosphosites identified in the rat synaptosomal preparation compared to that of the entire hippocampal mouse phosphoproteome may be due to the synaptic enrichment of the subcellular rat synaptosomal phosphoproteome or to technical advances in phosphoproteome coverage. Given these

**Fig. 1 | The recycling pool of SVs exhibit higher mobility than the total pool.**
**a** SdTIM of VAMP2-pHluorin-bound At647N-GBP nanobodies in SVs, indicative of recycling pool SV mobility. (i) Epifluorescence image of a neuronal segment expressing VAMP2-pHluorin acquired before incubation with At647N-GBP. Inset (red outline) highlights a presynaptic compartment, shown at higher magnification in (ii). (iii) Fluorescence intensity, (iv) diffusion coefficient (the color bar represents $\log_{10}[\mu m^2 s^{-1}]$) and (v) trajectory maps of recycling SVs. **b** SptPALM of vGLUT1-mEos2-containing vesicles, indicative of the total pool SV mobility. (i) Epifluorescence image of a neuronal segment expressing vGLUT1-mEos2. Inset (red outline) highlights a presynaptic compartment, shown at higher magnification in (ii). (iii) Fluorescence intensity, (iv) diffusion coefficient (the color bar represents $\log_{10}[\mu m^2 s^{-1}]$) and (v) trajectory maps of total SVs. **c**, Average MSD of VAMP2-pHluorin/At647N-GBP trajectories (Recycling pool; black), and vGLUT1-mEos2 (Total pool; red) as a function of time. **d** Area under the MSD curve (AUC; $\mu m^2$ s). **e** Frequency distribution of the diffusion coefficients [D] shown in a semi-log plot. Grey dashed line indicates the threshold used to distinguish the immobile ($\log_{10}[D]$

$\leq -1.6$) from the mobile ($\log_{10}[D] > -1.6$) fraction of molecules. **f** Plot of the mobile fraction of molecules. **g** Three-state model of diffusive states inferred by vbSPT analysis of VAMP2-pHluorin/At647N-GBP trajectories. **h**, Three-state model of diffusive states inferred by vbSPT analysis of vGLUT1-mEos2 trajectories. Circles in (**g**) and (**h**) represent diffusive states, where (D) is the diffusion coefficient. Immobile state (State 1), confined state (State 2) and highly mobile state (State 3). The areas of the circles represent the average state occupancy (%) of SVs in their respective states. The arrows indicate the transition probability of an SV moving from one state to the other. **i** Example of a trajectory from a recycling SV undergoing stochastic switching between the three diffusive states inferred by vbSPT analysis. Data in (**c**–**f**) are displayed as mean ± SEM. Values were obtained from $n = 15$ neurons (Recycling pool), and $n = 16$ neurons (Total pool), from over 3 independent neuronal cultures. 131 presynapses were analyzed in (**g**) and 35 presynapses were analyzed in (**h**). Statistical comparisons in (**d**, **f**) were performed using unpaired two-tailed Student's *t*-test with Welch's correction. Source data are provided as a Source Data file.

results, we next devised an unbiased correlative approach to identify highly intrinsically disordered proteins that undergo activity-dependent phosphorylation. For each presynaptic protein, the phosphopeptide with the highest upregulated phosphorylation value was used to assign a maximum $\log_2$ fold change (Fig. 2c). Candidate proteins were then identified by comparing the percentage of disorder with the activity-dependent increase in phosphorylation level for the mouse phosphoproteome (Fig. 2c, Supplementary Data 1). We refined our analysis by including the phosphoproteome data from stimulated rat synaptosomes[44] (Supplementary Fig. S2b, Supplementary Data 1). Among the identified candidate proteins, Tau, Bassoon, Synapsin 1, β-synuclein, Myristoylated Alanine Rich C-Kinase Substrate (MARCKS) and MARCKS-like protein (MLP) were some of the most highly disordered presynaptic proteins that also exhibited significant upregulation of phosphorylation in response to stimulation. Tau primarily works as an axonal protein controlling the transport of vesicles[24]. However, its function within synaptic terminals remains largely unexplored. Natively, Tau is an intrinsically disordered protein that, like Synapsin 1[13], has been shown to form biomolecular condensates in vitro[46] and in situ[18] within neurons. Tau is phosphorylated in an activity-dependent manner[44], regulating its association with microtubules[29] and the formation of biomolecular condensates[46]. Furthermore, our analysis showed that >90% of all mouse Tau phosphosites were identified between mouse and rat synaptosomes, of which >50% demonstrated conserved phosphorylation directionality in response to stimulation, including many of the most differentially phosphorylated phosphopeptides (Fig. 2b and Supplementary Fig. S2c; a full map of each mouse Tau phosphosite, phosphorylation status in rat synaptosomes and the corresponding conserved residue in the human Tau 2N4R isoform is provided in Supplementary Fig. S3). The mouse hippocampal phosphoproteome data exhibited the same phosphorylation site changes as presynaptic proteins known to undergo activity-dependent phosphorylation, such as the decreased phosphorylation of dynamin 1 at S774, and synapsin 1 at S62, as well as multiple increases and decreases of tau phosphorylation levels[47,44] (Fig. 2d and Supplementary Fig. S2c). Tau provided 53 peptides whose phosphorylation is regulated (up or down) in response to different types of stimulation both in neurons and synaptosomes (Supplementary Fig. S3). This resulted in the identification of 14 amino acids significantly phosphorylated in both datasets (Supplementary Fig. S2c). The activity-dependent phosphorylation of Tau, its ability to form biomolecular condensates, and its unknown role during synaptic transmission made Tau a strong candidate for investigation. We, therefore, performed sdTIM analysis on VAMP2-pHluorin-expressing hippocampal neurons from wild-type (WT) and Tau knockout (Tau KO) mice (Fig. 3a, b). Our results demonstrate that recycling SVs of Tau KO neurons were significantly more mobile than in WT neurons (Fig. 3c, d). We conducted a similar experiment to assess the mobility of the

total SV pool by performing sptPALM on vGLUT1-mEos2-transfected neurons from WT and Tau KO mice, which revealed no differences (Fig. 3e, f). These findings suggest that Tau explicitly controls the mobility of the recycling pool of SVs. To investigate whether Tau was sufficient to promote this effect, we performed rescue experiments (Fig. 3g, h). Tau re-expression in Tau KO neurons was sufficient to rescue the mobility of recycling SVs, which had been increased by the Tau KO (Fig. 3i, j). We next investigated the influence of phosphorylation status on SV mobility. Previous reports have suggested that SV mobility is regulated by synaptic phosphorylation states[48], with the phosphatase inhibitor okadaic acid (OkAc) dissipating SV clusters[11], and the protein kinase blocker staurosporine (Staur) increasing their clustering[12]. Therefore, we tested the effect of these two drugs on the mobility of individual vesicles within recycling (Fig. 3k, l) and total (Supplementary Fig. S4a, b) SV pools. As anticipated, OkAc-induced synaptic phosphorylation significantly increased the mobility of the recycling pool, and Staur-induced synaptic dephosphorylation promoted a marked decrease in the mobility of this pool (Fig. 3l), suggesting that dephosphorylation is crucial in conferring SV immobilization. The mobility of the total pool was also increased by OkAc treatment but was not affected by Staur (Supplementary Fig. S4a, b). Our results confirm that synaptic protein phosphorylation is a key regulator of the mobility of both the recycling and the total pool of SVs. We further tested whether the absence of Tau could prevent the phosphorylation-dependent increase in recycling SV mobility. We found that the previously observed OkAc-induced increase in recycling vesicle mobility was not observed in Tau KO neurons (Fig. 3m, n), reinforcing the notion that Tau is sufficient to control the mobility of the recycling SVs in response to phosphorylation. However, the absence of Tau did not prevent the decrease in recycling vesicle mobility induced by Staur (Fig. 3m, n). To assess the impact of Tau KO on the phosphorylation- and dephosphorylation-induced mobility changes in the total pool of vesicles, we performed sptPALM experiments in vGLUT1-mEos2-expressing neurons. Contrary to our results on the recycling pool, we found that OkAc promoted an increase in the mobility of the total pool of SVs in Tau KO neurons similar to that seen in WT neurons (Supplementary Fig. S4). Further, the low mobility of this pool was slightly decreased by Staur treatment (Supplementary Fig. S4c, d), confirming that other proteins may control the phospho-dependent mobility of the total pool of SVs, of which the majority comprise the reserve pool.

Taken together, our results indicate that Tau selectively controls the nanoscale organization of the recycling pool of SVs. Next, we investigated the nanoscale organization of Tau in axons and nerve terminals by performing sptPALM on neurons expressing Tau-mEos2 (Supplementary Fig. S5a), which revealed significant differences in the mobility between these neuronal compartments, with an evident immobilization of Tau within the presynapses (Supplementary

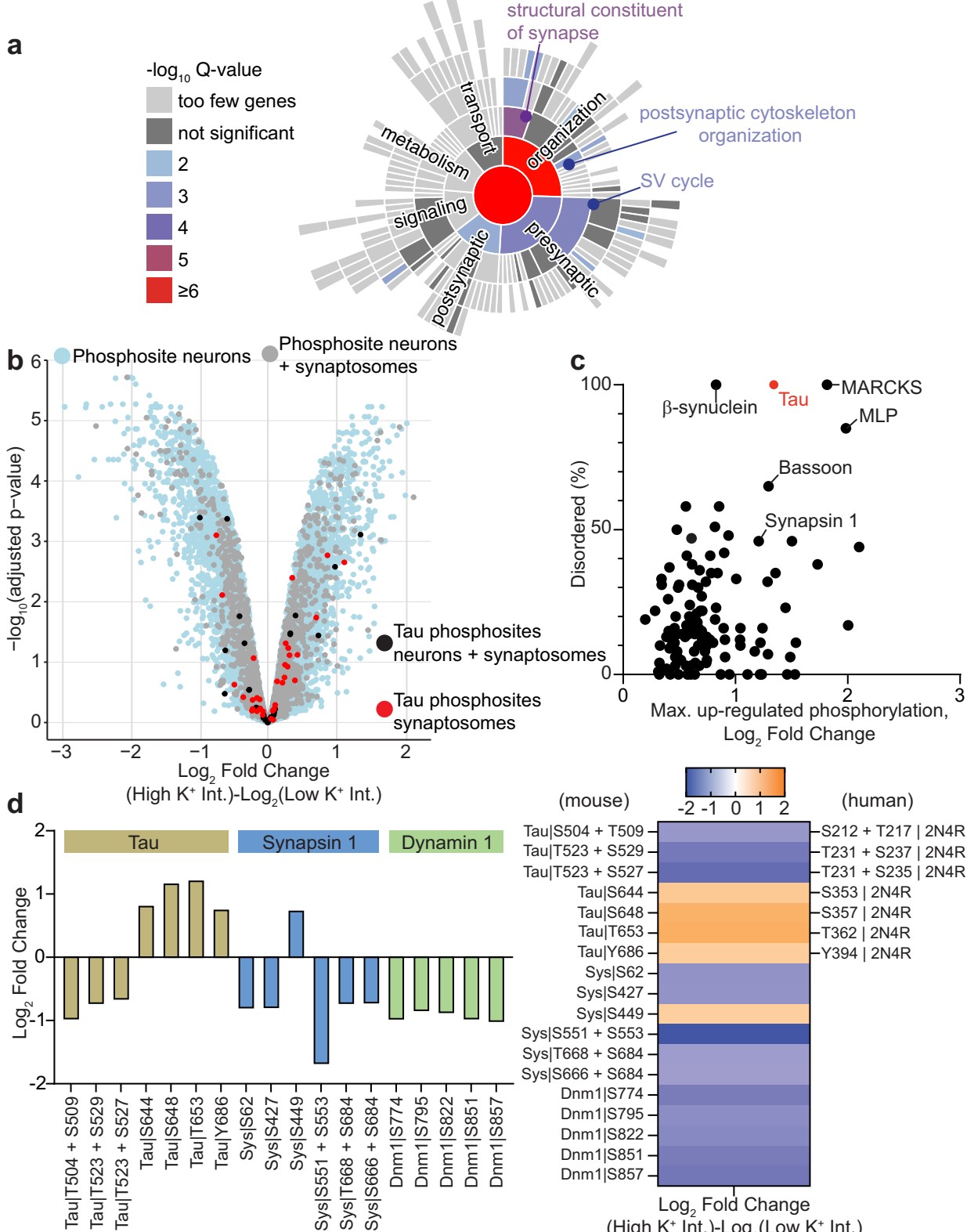

Fig. S5b, c). Pharmacological modification of Tau phosphorylation status showed a significant increase in the mobility of Tau proteins upon OkAc-induced phosphorylation. However, Staur treatment did not affect Tau mobility (Supplementary Fig. S6), reinforcing the idea that phosphorylation promotes the increase of Tau mobility observed during stimulation. To analyze whether Tau molecules form spatio-temporal clusters at the presynapse, we used Nanoscale

Spatiotemporal Indexing Clustering of trajectory segments (segNASTIC)[49]. This revealed that Tau forms nanoclusters at the pre-synapse and larger clusters (some appearing elongated) in the axonal compartment (Fig. 4a, panels i and ii). These nanoclusters were detected iteratively with an apparent lifetime of 10–20 s (Fig. 4b, panels i and ii, and Supplementary Fig. S7). Tau clustering behavior was different in the presynapse compared to the axon, with clustered

**Fig. 2 | Phosphoproteome of stimulated mature hippocampal neurons.** Mouse hippocampal neurons were stimulated (high K⁺) or resting (low K⁺) for 5 min and the phosphoproteome was analyzed by mass spectrometry. **a** SynGO (biological processes) analysis of the top 500 differentially phosphorylated proteins. **b** Volcano plot of all phosphopeptides identified in stimulated versus resting mouse hippocampal neurons. *X*-axis is the $\log_2$-ratio of high versus low phospho-peptide intensities and the *Y*-axis is the -$\log_{10}$ transformed adjusted p-value for the $\log_2$-ratios. Synaptosome phosphosites identified following global alignment of mouse and rat phosphosites is also superimposed (light blue indicates all mouse phosphopeptides; grey indicates phosphopeptides also identified in Rat; red indicates conserved Tau phosphopeptides; black indicates Tau sites not identified in

rat). **c** Each presynaptic protein is plotted based on the percentage of disordered sequence versus the maximum change in upregulated phosphorylation following stimulation. Tau (red) was selected as a candidate for further investigation. **d** $\log_2$ fold change for phosphorylation sites from known presynaptic activity-dependent phosphoproteins, Tau, synapsin 1 (Sys) and dynamin 1 (Dnm1). Tau conserved residues in the human 2N4R isoform are also specified. The mass spectrometry proteomics data and MaxQuant output have been deposited to PRIDE (PXD020232 and 10.6019/PXD020232). The adjusted *p*-value in (**b**) is the result of a moderated two-tailed Student's *t*-test corrected for multiple hypotheses using the Benjamini and Hochberg method (see Methods and Supplementary Data 1 for additional information). Source data are provided as a Source Data file.

molecules being less mobile and of higher density in presynapses (Fig. 4c, d). Further, the number of trajectories trapped within these clusters (cluster membership) (Fig. 4e), their apparent duration (cluster lifetime) (Fig. 4f), their area (Fig. 4g) and the density of molecules within these nanostructures (Fig. 4h) were all shown to be reduced in presynapses compared to axons. These results suggest that Tau clusters are differentially regulated in presynapses and axons. Given the known ability of Tau to form large biomolecular condensates in vitro[18,46], which are sensitive to protein concentration and perturbations of multivalent low-affinity interactions, we explored the sensitivity of Tau nanoclusters to these alterations. We fused Tau to Green Fluorescent Protein (GFP) and expressed it in human embryonic kidney (HEK-293T) cells. Increasing the expression levels of Tau-GFP resulted in the augmented formation of macroscopic Tau droplets (Supplementary Fig. S8a–c). As expected for a biomolecular condensate, fluorescence recovery after photobleaching (FRAP) analysis of Tau-GFP within these droplets showed a slower recovery compared with the cytosolic and microtubule-associated Tau, indicating slower molecular intermixing with surrounding Tau (Supplementary Fig. S8d, e). Importantly, these structures do not colocalize with markers of membrane-bound organelles, such as endosomes, the endoplasmic reticulum, or lysosomes (Supplementary Fig. S8f–h, respectively).

The aliphatic alcohol 1,6-hexanediol (1,6-HD) has previously been shown to inhibit the weak interactions between highly disordered protein domains that underpin the formation of biomolecular condensates[46]. Incubation of HEK-293T cells with 1,6-HD dissolved Tau-GFP droplets as evidenced by their reduced size and fluorescence intensity (Supplementary Fig. S9a–c). The ability of Tau to undergo LLPS depends on the presence of highly charged amino acid domains that facilitate and stabilize inter- and intramolecular interactions[50]. The microtubule-binding region (MTBR) of Tau is important in controlling Tau LLPS formation[46,51]. Two isoleucine-to-proline mutations within the MTBR of Tau (I277P and I308P) were shown to reverse the aggregation and LLPS formation phenotype of the "pro-aggregation" frontotemporal dementia ΔK280 mutant[18,52,53]. Introducing the two proline mutations into Tau-GFP (Tau^I277P / I308P-GFP) (Supplementary Fig. S9d) resulted in a significant reduction in the number of cells exhibiting Tau LLPS droplets and in the number of Tau LLPS droplets per cell (Supplementary Fig. S9e, f). The use of a Tau construct lacking the MTBR and C-terminal regions (ΔTau74-GFP)[51,54] (Supplementary Fig. S9d) further reduced the number of cells harboring Tau LLPS droplets and the number of Tau LLPS droplets per cell, as well as significantly decreasing their size (Supplementary Fig. S9e–g). These results confirm that these Tau droplets are large biomolecular condensates[18]. Tau condensates have also been observed in neurons[18]. After expressing Tau-GFP in Tau KO hippocampal neurons, we could also detect the presence of Tau condensates (Supplementary Fig. S10a, b). FRAP analysis of these condensates showed slow recovery after photobleaching and a significantly reduced mobile fraction compared to cytosolic Tau-GFP (Supplementary Fig. S10c–e). Similar to our observations in HEK-293T cells, neuronal Tau condensates did not colocalize with endosomes, the endoplasmic reticulum or lysosomes (Supplementary Fig. S10f–h, respectively). These results confirm the

membrane-less state of Tau droplets, further supporting the idea that these structures are indeed biomolecular condensates.

To evaluate the effect of protein levels on the nanoscale organization of Tau, we compared the clustering of Tau-mEos2 re-expressed in Tau KO neurons (Supplementary Fig. S11a) with that of neurons from Tau TALEN-based gene-edited mice (Supplementary Fig. S11b), in which Tau-mEos2 endogenous expression level is ~15-times lower than wild-type Tau[55]. As expected, significantly fewer trajectories were detected in TALEN mice-derived neurons (Supplementary Fig. S11c). Cluster analysis on Tau KO neurons re-expressing Tau-mEos2 and on Tau-mEos2 TALEN neurons, further revealed a decrease in cluster membership, lifetime, area and in the immobilization of Tau molecules within the axonal compartment (Supplementary Fig. S11d–g). However, there were no significant changes in these clustering metrics at the presynapse (Supplementary Fig. S11h–l). We next used 1,6-HD in live neurons expressing Tau-mEos2 to assess the sensitivity of Tau's nanoscale organization to this LLPS inhibitor (Fig. 5a, b). Incubation with 1,6-HD increased the mobility of Tau-mEos2 both at the presynapse (Fig. 5c, d) and the axonal compartment (Fig. 5e, f). To further evaluate how Tau's LLPS influences its organization at the nanoscale level, we generated super-resolution-compatible Tau constructs with reduced LLPS ability. Expression of Tau^I277P / I308P-mEos2 (Fig. 5g, panel ii) in the Tau KO background showed significantly increased mobility both at the presynapse and the axon (Fig. 5h–k), and ΔTau74-mEos2 (Fig. 5g, iii) had further increased mobility in the axon (Fig. 5h–k). Cluster analysis confirmed that both Tau LLPS-deficient mutants showed increased Tau mobility within clusters, a reduced percentage of clustered trajectories and a reduced density of clustered molecules (Supplementary Fig. S12). Taken together, these data show that Tau forms synaptic and axonal nano-biomolecular condensates that are sensitive to 1,6-HD and LLPS-reducing mutations.

We next tested whether endogenous Tau localization was affected by synaptic activity. The Tau-5 antibody recognizes residues located at the proline-rich domain region of Tau regardless of its phosphorylation state, allowing the detection of total Tau levels (phosphorylated and non-phosphorylated). Stimulated and resting neurons were fixed and immunostained with Tau-5 antibody (Supplementary Fig. S13a, b), revealing a significant increase in anti-Tau-5 fluorescence intensity at presynapses (as indicated by VAMP2 staining) in response to stimulation (Supplementary Fig. S13c). To visualize active nerve terminals, we performed sptPALM with Tau-mEos2 in live neurons co-expressing VAMP2-pHluorin (Fig. 6a). Depolarization with high K⁺ resulted in an evident redistribution of Tau molecules to unquenched VAMP2-pHluorin-positive areas, indicative of presynaptic release sites (Fig. 6a). We found a significant increase in the number of Tau-mEos2 trajectories detected at the presynapse, confirming the activity-dependent redistribution of Tau to the presynapse in live neurons (Fig. 6b). Concomitantly, we found a decrease in the number of trajectories within neighboring axonal segments, further suggesting that presynaptically-detected Tau originates from the axon (Fig. 6c). Consistent with this axo-terminal diffusion, analysis of Tau-mEos2 mobility revealed an activity-dependent increase in the presynapse (Fig. 6d, e) and a decrease in neighbouring axonal segments (Fig. 6f, g). This

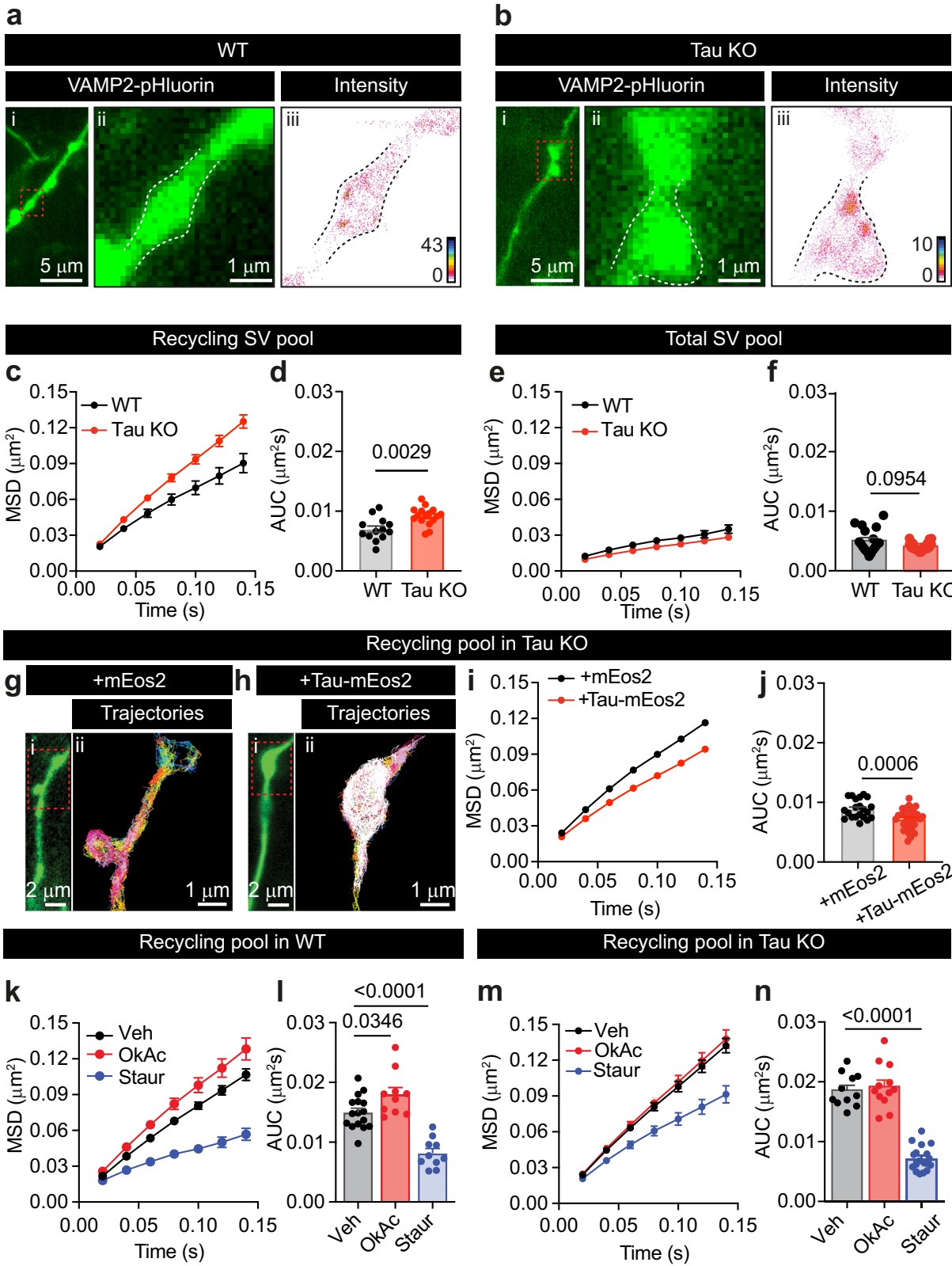

suggests that Tau molecules, which are known to form fast kiss-and-hop interactions with microtubules[28], could diffuse from the abutting axon and subsequently accumulate at the presynapse in response to stimulation. We next asked whether translocated Tau could intermix with recycling vesicles at the stimulated presynapses. To simultaneously visualize single Tau molecules and individual recycling SVs in live hippocampal neurons, we performed dual-color single-molecule

tracking of Tau-mEos2 using sptPALM and of recycling vesicles using sdTIM (Supplementary Video S3). In resting nerve terminals, we detected low numbers of Tau molecules in the area also occupied by recycling SVs (Fig. 6h). Importantly, in response to stimulation, Tau molecules largely intermixed with recycling vesicles occupying the presynapse (Fig. 6i). Interestingly, stimulation also affected Tau presynaptic nanocluster behavior (Fig. 6j, k). Upon stimulation, both the

**Fig. 3 | Tau controls the mobility of the recycling pool of SVs. a**, **b** (i) Epifluorescence image of VAMP2-pHluorin expressed in WT neurons (**a**) or Tau KO neurons (**b**) incubated with At647N-GBP nanobodies. Insets (red outline) on a presynapses are shown at higher magnification in (ii). (iii) Intensity maps of recycling SVs. **c**, **e** Average MSDs of recycling SVs (**c**) or total SVs (**e**) from WT (black) or Tau KO (red) neurons. **d**, **f** Plots of the area under the MSD curves (AUC). **g** (i) Epifluorescence image VAMP2-pHluorin expressed in Tau KO neurons incubated with At647N-GBP nanobodies. Empty mEos2 was expressed as a control. Inset (red outline) on a presynapse is shown at higher magnification in (ii) (trajectory map of SVs). **h** (i) Epifluorescence image of VAMP2-pHluorin expressed in Tau KO neurons where Tau-mEos2 was re-expressed, and incubated with At647N-GBP nanobodies. Inset (red outline) on a presynapse is shown at higher magnification in (ii) (trajectory map of SVs). **i** Average MSD of recycling SVs from neurons expressing mEos2 (black) or Tau-mEos2 (red). **j** The corresponding area under the MSD curves (AUC).

**k** Average MSD of recycling SVs from Tau WT neurons treated with control vehicle (Veh, DMSO; black), okadaic acid (1 µM, 15 min, OkAc; red), or staurosporine (1 µM, 15 min, Staur; blue). **l** The corresponding area under the MSD curves (AUC). **m** Average MSD of recycling SVs from Tau KO neurons treated with control vehicle (Veh, DMSO; black), okadaic acid (OkAc; red), or staurosporine (Staur; blue). **n** The corresponding area under the MSD curves (AUC). Data are displayed as mean ± SEM. Values were obtained from $n = 13$ and 16 neurons in (**c**, **d**), $n = 17$ and 18 neurons in (**e**, **f**), $n = 21$ and 31 neurons in (**i**, **j**), $n = 16$, 10 and 10 neurons in (**k**, **l**), and $n = 11$, 12 and 18 neurons in (**m**, **n**). Data was obtained from >2 independent neuronal cultures. Statistical comparisons were performed using the unpaired two-tailed Student's $t$-test in (**d**) and (**f**) and the unpaired two-tailed Mann-Whitney $U$ test in (**j**), and using the one-way ANOVA test followed by Dunnett post hoc test comparing the groups to the control (Veh) in (**l**) and (**n**). Source data are provided as a Source Data file.

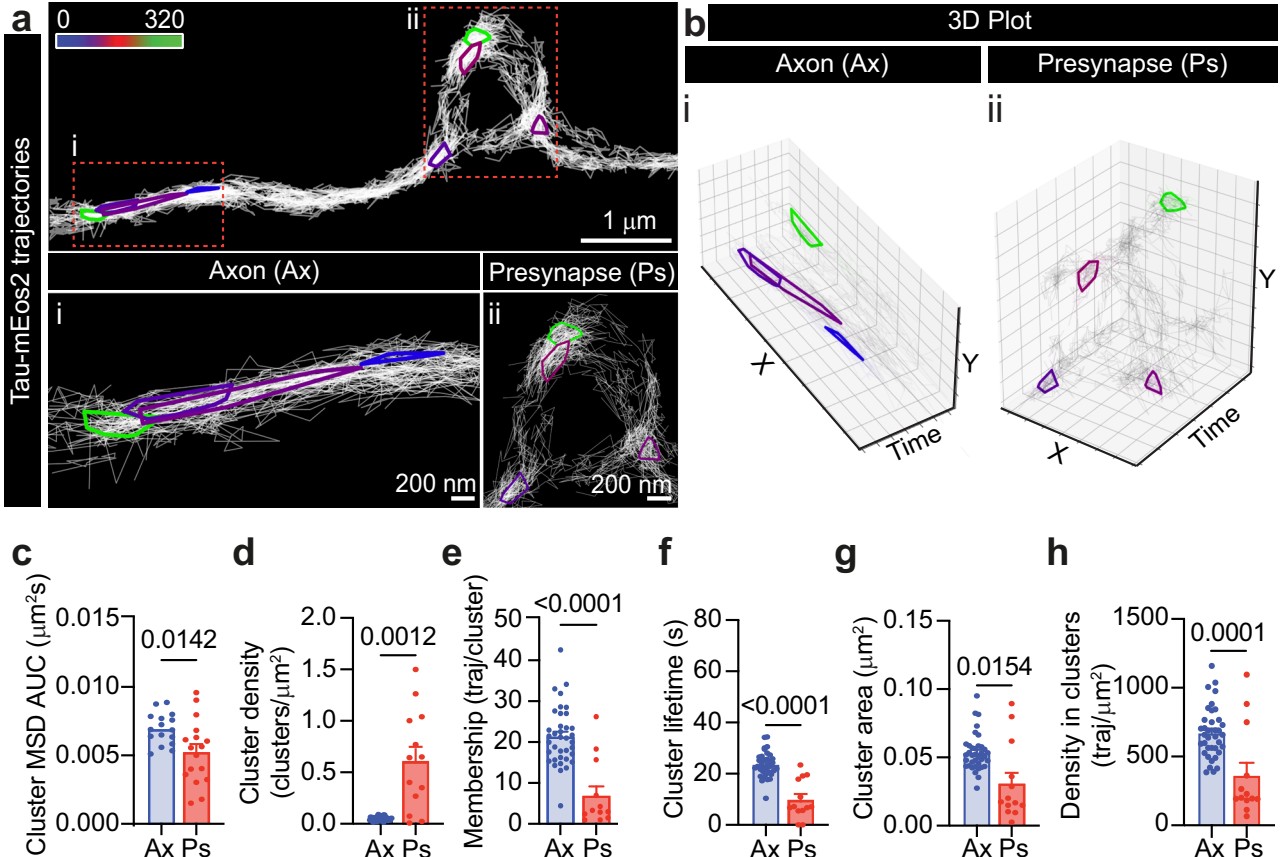

**Fig. 4 | Tau forms differently regulated presynaptic and axonal nanoclusters. a** Representative Tau-mEos2 trajectories generated using segNASTIC showing nanoclusters at the presynapse and in the axonal compartment. Color-coding of the clusters represents their appearance in time across the acquisition (16,000 frames, 320 s). The insets (red outlines) of the (i) axonal or (ii) presynaptic compartment are shown at a higher magnification. **b** (i) 3D (X, Y, Time) plot of Tau-mEos2 trajectories from the axonal compartment. (ii) 3D (X, Y, Time) plot of Tau-mEos2 trajectories from the presynaptic compartment. Squares in X represent 200 nm; squares in Y represent 100 nm in (i) or 200 nm in (ii); squares in Time represent 50 s. **c–h** Comparison of cluster metrics from the axonal (Ax) and the presynaptic (Ps)

compartments. **c** Average MSD of clustered Tau-mEos2 trajectories represented as the area under the curve (AUC). **d** Average density of Tau-mEos2 clusters per µm². **e** Average cluster membership (trajectories/cluster). **f** Average apparent lifetime of Tau-mEos2 clusters. **g** Average Tau-mEos2 cluster area. **h** Average density of Tau-mEos2 trajectories per cluster area. Data are displayed as mean ± SEM. Values were obtained from $n = 17$ synapses in (**c**) or 13 synapses in (**d**) to (**h**), and $n = 15$ axons in (**c**) or 37 axons in (**d**) to (**h**). Data was obtained from ≥2 independent neuronal cultures. Statistical comparisons were performed using the unpaired two-tailed Student's $t$-test with Welch's correction. Source data are provided as a Source Data file.

percentage of Tau clustered trajectories and the density of synaptic clusters significantly increased (Fig. 6l, m), with little change to the other cluster metrics (Fig. 6n, o).

## Discussion

Despite distinct release probabilities and mobilities, the reserve and recycling vesicles co-exist within a single cluster at the presynapse. To

start unravelling this conundrum, we conducted a large-scale phosphoproteomics analysis to identify Tau as a candidate to form synaptic biomolecular condensates that might regulate the clustering of the SVs. Using single-molecule super-resolution microscopy, we discovered that Tau undergoes liquid-liquid phase separation, forming transient presynaptic nanoclusters whose density and number are regulated by synaptic activity. We revealed that this activity-dependent

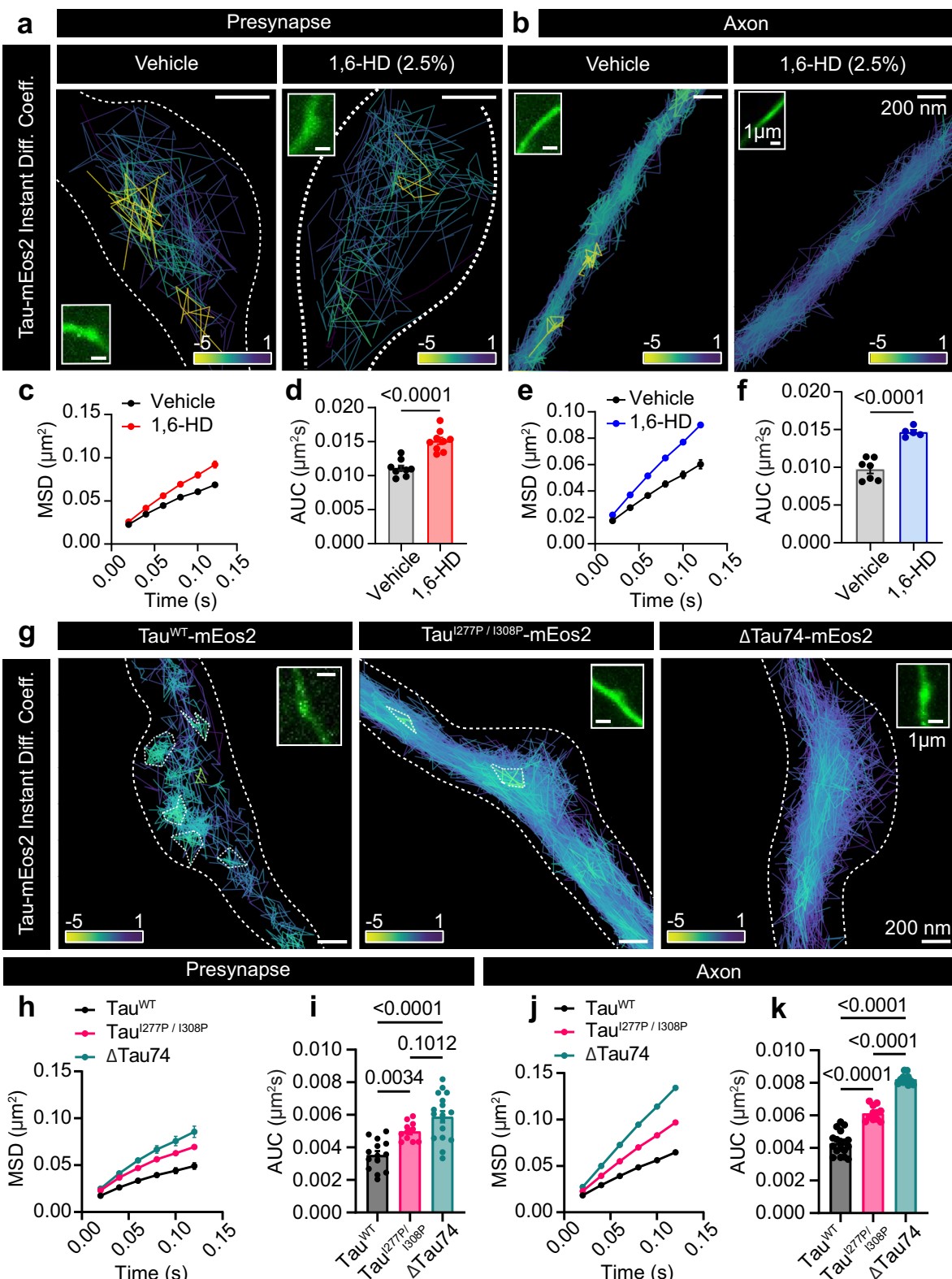

process allows Tau to diffuse into the presynapse, creating nanoscale biomolecular condensates that selectively control the mobility of recycling vesicles.

We first implemented a set of single-molecule imaging techniques to track individual SVs from distinct pools in live mouse hippocampal presynapses and compare their nanoscale organization and mobility. To track individual SVs of the recycling pool, we used sdTIM[9,10], which

relies on the internalization of fluorescently-tagged anti-GFP nanobodies bound to VAMP2-pHluorin molecules. This approach provided metrics that are in good agreement with those previously reported in rat neurons[9,10]. We used vGLUT1-mEos2, a prototypical vesicular protein widely used to image SVs in central synapses[35], to track individual vesicles of the total pool. As 90% of the total pool of SVs belongs to the reserve pool[5], metrics stemming from vGLUT1 sptPALM experiments

**Fig. 5 | Tau mobility is sensitive to 1,6-HD. a, b** Representative Tau-mEos2 trajectories generated using segNASTIC showing instantaneous diffusion coefficients (the color bar represents $\log_{10}[\mu m^2 s^{-1}]$) in an (**a**) synaptic region and (**b**) an axonal region, determined by VAMP2-pHluorin intensity (inset), of neurons treated with control vehicle (DMSO) or with 1,6-HD (2.5%). **c** Average MSD of Tau-mEos2 presynaptic trajectories from neurons treated with control vehicle or with 1,6-HD. **d** Corresponding areas under the MSD curves (AUC). **e** Average MSD of Tau-mEos2 axonal trajectories from neurons treated with control vehicle or with 1,6-HD. **f** Corresponding areas under the MSD curves (AUC). **g** Representative segNASTIC trajectories showing instantaneous diffusion coefficients from Tau-mEos2, and Tau LLPS mutants Tau$^{I277P/I308P}$-mEos2 and ΔTau74-mEos2. **h, j** Average MSD of the trajectories of mEos2-tagged Tau WT and LLPS Tau mutants in (**h**) presynapses and

(**j**) axons. **i, k** Corresponding areas under the curve (AUC), in the (**i**) presynapses and (**k**) axonal compartment. Data in (**c**) to (**f**) and (**h**) to (**k**) are displayed as mean ± SEM. Data in (**c, d**) was obtained from $n = 9$ synapses from 9 different neurons. Data in (**e, f**) was obtained from $n = 7$ and 5 axons from different neurons. Data in (**h**) to (**k**) was obtained from $n = 14$ presynapses from 14 neurons and 18 axons from 18 neurons in Tau$^{WT}$, $n = 11$ synapses and axons from 11 neurons in Tau$^{I277P/I308P}$, and $n = 17$ synapses from 17 neurons and 12 axons from 12 neurons in ΔTau74. Data was obtained from 2 independent neuronal cultures. Statistical comparisons in (**d**) and (**f**) were performed using the unpaired two-tailed Student's *t*-test with Welch's correction in (**d**) and (**f**), or using the one-way ANOVA test in (**i**) and (**k**). Source data are provided as a Source Data file.

are a good indicator of the nanoscale organization of the reserve vesicular pool. As anticipated, these two pools revealed distinct behaviors, with a more mobile recycling pool compared to that of the total (reserve) pool (Fig. 7a). The heterogeneous nature of these two pools' mobilities was further analysed using variational Bayes single-particle tracking (vbSPT) analysis[37,40]. This analysis classically assigns a discrete number of hidden diffusive states and transitions between these states from experimental data using hidden Markov modelling. For both pools of SVs, the immobile state was similar in terms of diffusion coefficient, state occupancy, and probability of transitioning to the confined state. The two pools differed in the number of vesicles exhibiting confined and highly mobile states. Most SVs of the recycling pool were mobile, while the total pool had more vesicles in a confined state. Notably, the transition between immobile and mobile states occurs via the confined state for both pools. This suggests that the mechanism leading to the immobilisation of both pools may have some commonality and that highly mobile vesicles require at least a two-stage process to become immobile. The immobilisation of both pools is likely to relate to their known ability to cluster at the presynapse, a process that has been in the spotlight recently with the recent discovery that biomolecular condensates might be involved[13]. In the synaptic context, LLPS allows proteins with critical synaptic functions, such as endocytosis and exocytosis, to concentrate in a protein-dense phase separated from the surrounding milieu[56]. To identify proteins undergoing LLPS, it is crucial to assess their level of intrinsic disorder, which is known to be involved in biomolecular condensate formation.

Many synaptic proteins undergo activity-dependent phosphorylation and dephosphorylation[44,57]. To identify proteins involved in controlling the clustering of the recycling pool of SVs, we performed an unbiased large-scale phosphoproteomic analysis of stimulated hippocampal synapses. By comparing the activity-dependent phosphorylation level from mature hippocampal neurons and synaptosomes, we could analyze synaptic versus non-synaptic proteins and characterize their phosphorylation level. This was correlated with their degree of disorder to highlight candidates playing a role in forming synaptic biomolecular condensates. We found 4626 phosphopeptides that differed significantly in their phosphorylation status in response to stimulation. Among the candidate proteins identified by correlating the percentage disordered with maximum phosphopeptide upregulation for the mouse phosphoproteome, Tau, bassoon, synapsin-1, β-synuclein, Myristoylated Alanine Rich C-Kinase Substrate (MARCKS) and MARK-like protein (MLP) were the proteins with the highest values. While LLPS has been demonstrated for Tau and synapsin-1[13,46], no phase separation has been reported for bassoon, MARCKS and MLP. Disorder profile analysis predicted bassoon and β-synuclein as high-profile candidates to undergo LLPS[56,58]. β-Synuclein exhibits high sequence homology and structural similarity with α-synuclein, a protein implicated in Parkinson's Disease whose ability to undergo LLPS precedes its toxic aggregation[59]. However, recent evidence failed to demonstrate phase separation in β-synuclein[59].

Tau is predominantly expressed in the nervous system as an axonal protein but its function within synaptic terminals remained largely unexplored. We found that Tau KO selectively and significantly increased the mobility of the recycling pool of SVs (Fig. 7b). Re-expression of Tau in the Tau KO background decreased the mobility of the recycling SVs back to their normal levels. Tau is therefore necessary and sufficient to control the presynaptic clustering of the recycling pool of SVs. The mechanism by which Tau controls the recycling pool is currently unknown, but Tau was recently shown to bind the SV protein synaptogyrin-3, an effect critically involved in synaptic dysfunction associated with several tauopathies[60,61]. How Tau and synaptogyrin-3 control this pool selectively will require additional investigations, but it is tempting to hypothesise that such binding allows SVs to associate with a presynaptic biomolecular condensate in a phosphorylation-dependent manner likely regulated by synaptic activity. Tau is an intrinsically disordered protein that can form biomolecular condensates in vitro and when it is expressed in heterologous cellular systems[18] and neuronal cultures[51,62]. Tau mobility has been previously analysed in the context of axonal microtubule binding[28] and clustering[63], and its ability to immobilize the Fyn kinase at the postsynapse[51,62]. To the best of our knowledge, our study is the first to report the dynamic nanoscale organisation of Tau at the presynapse. We used single-molecule super-resolution microscopy to demonstrate that Tau can generate transient nanoclusters characterized by a series of spatiotemporal metrics such as cluster size, cluster density per presynapse and cluster apparent lifetime. Perisynaptic axonal Tau nanoclusters behave differently than presynaptic ones. Both the size of the presynaptic Tau nanoclusters and the number of Tau trajectories detected within these clusters are significantly smaller within nerve terminals. However, the density of Tau nanoclusters is much higher at the presynapses, and their apparent lifetime is shorter (Fig. 7c). These observations reveal the highly dynamic nature of Tau molecules, along with the regulation of their nanoclusters at the presynapse by synaptic activity. Axonal Tau nanoclusters, on the other hand, likely act as a reserve pool of Tau proteins that can dynamically release Tau molecules in an activity-dependent manner, allowing them to diffuse to the nearby presynapses (Fig. 7d) where they can interact with synaptogyrin-3 and associated SVs[60,61]. The activity-dependent switch in Tau localisation from the axon to nerve terminals likely depends on Tau phosphorylation status[64], which is known to regulate its association with microtubules and the formation of biomolecular condensates[18]. Assigning the formation of Tau nanoclusters to either microtubule binding or biomolecular condensates remains elusive. However, the use of two mutants (Tau$^{I277P/I308P}$ and ΔTau74) with distinct reduced abilities to form condensate[18,51] strongly advocate for a major involvement of phase separation in Tau nanocluster in both axons and synapses. Further, the Tau nanocluster sensitivity to 1,6-hexanediol adds to the evidence favouring Tau nanoscale biomolecular condensate formation in these regions. The mechanisms of conversion of monomeric proteins into biomolecular condensate droplets and their irreversible progression into fibrils are not entirely understood, but are of special relevance for several disorders, including Alzheimer's and Parkinson's disease[65]. Tau

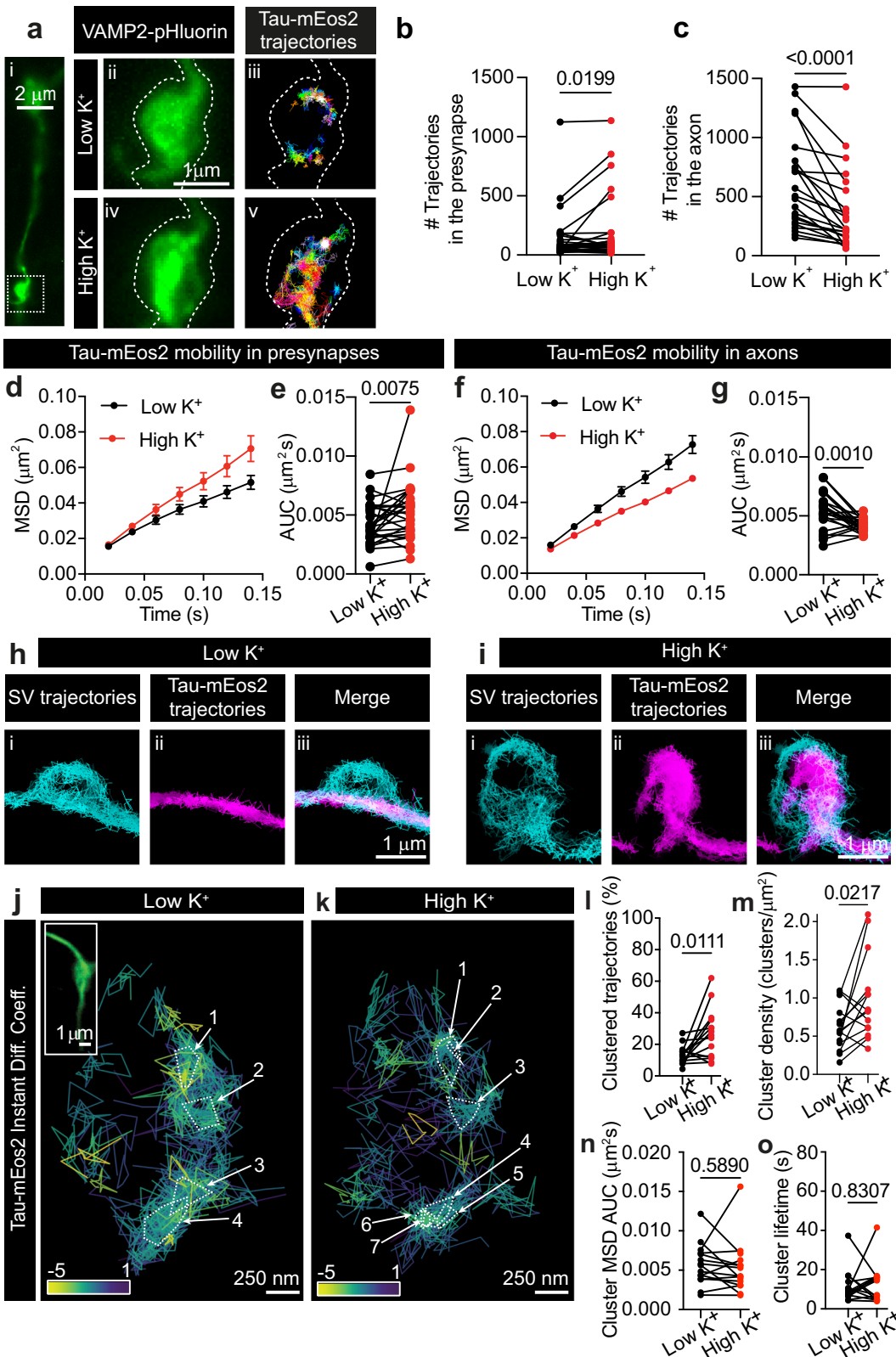

LLPS can work as nucleating seeds that accelerate a liquid-to-solid phase transition and the formation of toxic amyloid fibrils[18], and α-Synuclein can efficiently aggregate into amyloid fibrils inside condensate droplets[66,67]. Interestingly, macroscopic α-Synuclein phase-separated droplets originate at physiologically relevant concentrations from reversible liquid-like nanoscale clusters containing tens to hundreds of molecules[68]. Tau organises along microtubules in cell lines and neurons,

forming nanoclusters that precede toxic aggregate fibrils[63]. Monomeric, dimeric, and trimeric Tau complexes form these nanoclusters[63], but it is not known whether these subdiffractional structures are associated with Tau's ability to undergo phase separation. Our results indicate that Tau nanoclusters are associated with its organization as biomolecular condensates, highlighting the importance of a deeper understanding of Tau nanoscale phase separation to develop more efficient treatments

**Fig. 6 | Tau nanoscale organization is regulated by synaptic activity. a** (i) Representative epifluorescence image of a WT hippocampal neuronal segment expressing VAMP2-pHluorin and Tau-mEos2, used for imaging under resting conditions (Low K⁺) and after stimulation (High K⁺). (ii) The inset of the presynaptic compartment in resting condition is shown at a higher magnification. (iii) Trajectory map of Tau-mEos2 in the unstimulated presynaptic compartment. (iv) The inset of the presynaptic compartment during stimulation is shown at a higher magnification. (v) Trajectory map of Tau-mEos2 in the stimulated presynaptic compartment. **b** Plot showing the total number of Tau-mEos2 trajectories within the presynapse, in resting conditions (black) and after stimulation (red). **c** Plot showing the total number of Tau-mEos2 trajectories within the axon, in resting conditions (black) and after stimulation (red). **d–g** Average MSD of total Tau-mEos2 trajectories and the corresponding area under the curve (AUC) in resting conditions (black) and after stimulation (red), in the presynapse (**d, e**) and in the axonal compartment (**f, g**). **h** Trajectory map of (i) recycling SVs and (ii) Tau-mEos2 in the

unstimulated presynaptic compartment, obtained by dual-color single-tracking super-resolution imaging. (iii) Merged image. **i** Trajectory map of (i) recycling SVs and (ii) Tau-mEos2 in the stimulated presynaptic compartment, obtained by dual-color single-tracking super-resolution imaging. (iii) Merged image.

**j, k** Representative Tau-mEos2 trajectories generated using segNASTIC showing instantaneous diffusion coefficients (the color bar represents $\log_{10}[\mu m^2 s^{-1}]$) in a presynaptic region, determined by VAMP2-pHluorin intensity (inset), of a neuron that was unstimulated (**j**) and subsequently stimulated (**k**). **l** Percentage of Tau-mEos2 trajectories that are clustered. **m** Average cluster density of Tau-mEos2. **n** Average MSD of clustered Tau-mEos2 trajectories represented as the area under the curves (AUC). **o** Average apparent lifetime of Tau-mEos2 clusters. Data are displayed as mean ± SEM. Values were obtained from $n = 29$ synapses in (**b, e**), $n = 23$ axons in (**c, g**), and $n = 15$ synapses in (**l, m, n, o**). Data was obtained from ≥2 independent neuronal cultures. Statistical comparisons were performed using the two-tailed Student's paired *t*-test. Source data are provided as a Source Data file.

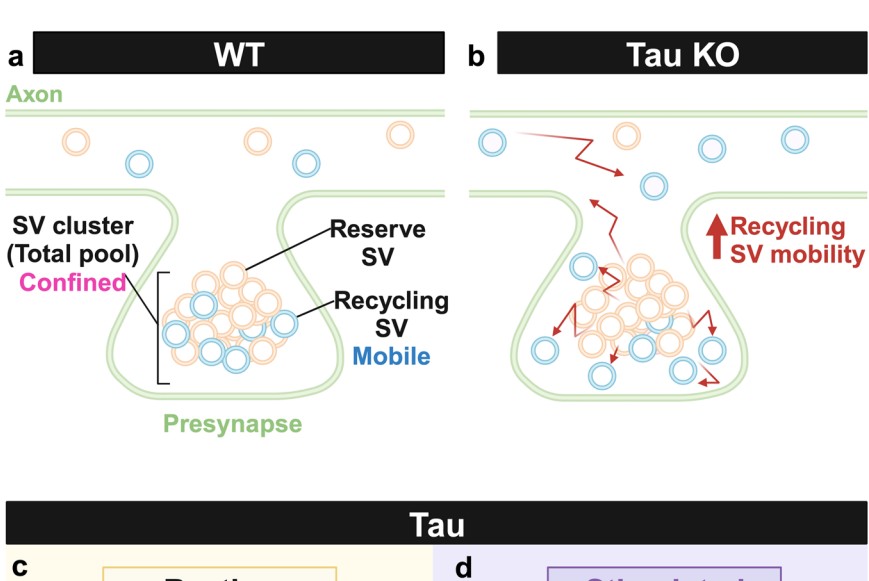

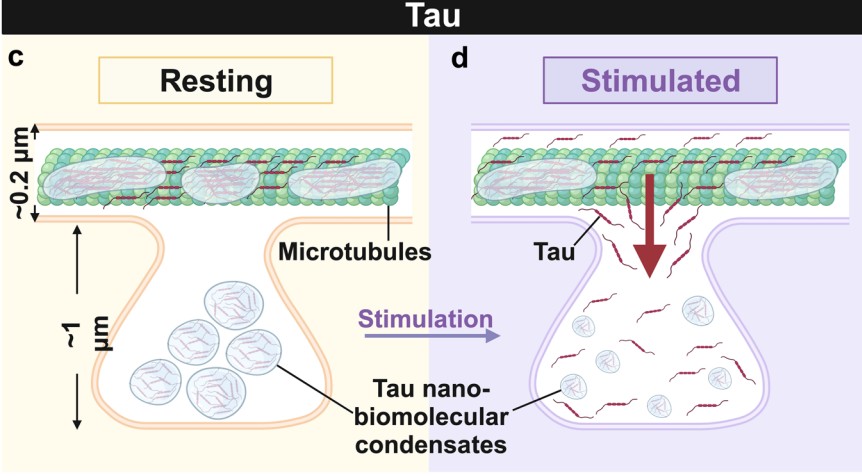

**Fig. 7 | Graphical representation of Tau nanoscale organization in axons and nerve terminals and its role in controlling the clustering the recycling pool of SVs. a, b** Graphical representation of SVs in WT (**a**) and Tau KO (**b**) nerve terminal neurons. Tau KO impacts the nanoscale organization of recycling SVs within the presynaptic bouton, resulting in an increase in their mobility. **c, d** Graphical representation of Tau nano-biomolecular condensates in nerve terminals in resting

(**a**) and stimulated (**b**) conditions. Under resting conditions, the axonal segment contains larger Tau nanoclusters, whereas synaptic boutons harbor a more dense population of smaller Tau nano-biomolecular condensates. Neuronal stimulation increases the number of Tau molecules detected at the presynapse and promotes a re-organization of Tau nano-biomolecular condensates, leading to more numerous, smaller Tau nano-biomolecular condensates. Created with BioRender.com.

for dementia-related disorders. Tau nanoclusters are dynamically regulated, and they can also alter the cluster organization of other proteins, such as Fyn in dendritic spines in disease conditions[62]. Indeed, mutant Tau associated with the development of Frontotemporal dementia can also modify Fyn clusters by forming phase condensates, immobilizing Fyn molecules into smaller but more crowded clusters[51].

Together, these observations suggest that both phosphorylation and phase separation are critical mechanisms Tau uses to dynamically regulate the organization of proteins into clusters in the nervous system.

Our study indicates that Tau selectively controls the nanoscale organization of the recycling pool of SVs. Individual recycling SVs from Tau KO neurons were significantly more mobile than WT neurons

(Fig. 7b). The total SV pool, which mainly contains reserve pool vesicles, was not affected in Tau KO mice, suggesting that Tau controls the mobility of the recycling pool of SVs. Tau re-expression in Tau KO neurons rescued the mobility of recycling SVs demonstrating that Tau is both necessary and sufficient to control recycling SVs mobility. Further, the use of okadaic acid, which promotes synaptic phosphorylation, allowed us to determine that the recycling vesicle pool is controlled by phosphorylation. Okadaic acid is a potent inhibitor of the protein phosphatase PP1 and the protein phosphatase 2 A (PP2A), which results in the activation of kinases such as MAPK, ERK, PKA, JNK, PKC and GSK3β, and the phosphorylation of Tau[69]. The lack of effect of okadaic acid[11,70] on recycling SVs from Tau KO neurons also advocates for a unique role of phospho-Tau in this process. Although okadaic acid lacks selectivity, our results suggest that Tau alone is critical for the clustering of recycling SVs. Importantly, staurosporine, which promotes synaptic protein dephosphorylation[12] by inhibiting kinases such as protein kinase C (PKC), protein kinase A (PKA), and protein kinase G (PKG)[71,72], dramatically reduces the mobility of the recycling and the reserve pool. The absence of Tau does not affect staurosporine-induced change in SV mobility. This indicates that dephosphorylation of Tau and other phosphoproteins is required for the release of SVs from their anchor/clusters. Additional molecular components must control the exit of SV from their molecular anchoring in a dephosphorylation-dependent manner.

The molecular mechanism underpinning the selectivity for the recycling pool is currently unknown but is likely to stem from the transient interaction of Tau with synaptogyrin-3[60,61]. However, synaptogyrin-3 is likely present in all pools of SVs. It is interesting to note that although recycling SVs have been shown to localize throughout the cluster of vesicles, they are usually found at or near the periphery of the volume of the SV cluster[8]. It is tempting to speculate that the rapid diffusion and transient clustering of Tau at the presynapse could help immobilise the recycling SVs at the outskirt of the SV cluster. However, more work will be needed to address these crucial points. Further, hippocampal synapses have well-characterised "en passant" presynapses, and the geometry of the access points for tau diffusion may be slightly different in "terminal" presynapses as only one route of entry is present instead of a continuum of the perisynaptic axon. This could alter the kinetic of diffusion and, ultimately, the timing of presynaptic clustering. More work will be needed to address these crucial points.

Synapsin-1 was one of the first examples of a presynaptic protein capable of undergoing LLPS[13] in vitro. This seminal discovery suggested the synapsin-1-dependent LLPS model for SV clustering as a complementation of the classical synapsin-SV-actin tethering model[73]. However, recent findings have shown the existence of different mechanisms of synapsin-induced vesicle clustering[74,75]. Synapsin-1 function has been directly linked to efficiently maintaining SVs within this reserve pool[76]. However, the mechanism allowing the clustering of the recycling pool of SVs has been elusive. Our findings demonstrate that Tau forms transient nano-biomolecular condensates and plays a key role in regulating the mobility of the recycling pool of SVs, with Tau phosphorylation being critical in this modulation. We have identified an axonal population of Tau nanoclusters with different dynamic properties from presynaptic Tau. Furthermore, we have also demonstrated that neuronal stimulation triggers Tau phosphorylation, which initiates the redistribution of Tau molecules from the axonal microtubule-enriched compartment into adjacent presynapses, which are largely devoid of microtubules. We propose that Tau presynaptic influx results in the generation and remodelling of existing Tau nano-biomolecular condensates at the synapses, which then intermix with SVs to control their nanoscale organization during neuronal communication.

Our results provide new insights into how Tau forms nanoscale biomolecular condensates that selectively regulate the dynamic organization of recycling SVs, an effect controlled by Tau phosphorylation status. We have revealed a novel mechanism through which synaptic activity finely controls the nanoscale organization of the Alzheimer's disease protein Tau into biomolecular condensates. Dysregulation of these transient synaptic nano-biomolecular condensates may alter the axonal specificity of Tau by facilitating its diffusion towards the somatodendritic compartment[77,78]. This dysregulation may also produce "seeds" that generate large phase-separated structures that can further aggregate to form amyloid fibrils, a hallmark of Alzheimer's disease. Our study, therefore, has important ramifications for the current understanding of the mechanisms underlying synaptic transmission and for deriving new treatments for tauopathies. Our findings unequivocally demonstrate the existence of nanoscale biomolecular condensates at the synapse. Therefore, the large cellular biomolecular condensates may solely represent the macroscopic manifestation of a much broader, dynamic and likely functionally important phenomenon at the nanoscale level.

## Methods

### Ethics and inclusion statement
Most of the research has been performed including local researchers from the University of Queensland and the University of Sydney. Roles and responsibilities were agreed upon amongst all the collaborators, and we have not discriminated against any individual based on gender, race, age, religion, sexual orientation, disability status or veteran status. All experimental procedures using animals were conducted under the guidelines of the Australian Code of Practice for the Care and Use of Animals for Scientific purposes and were approved by the University of Queensland Animal Ethics Committee (2020/AE000439; 2020/AE000204).

### Mouse strains and neuronal culture
Mice were maintained in a 12-hr light/dark cycle (80% intensity), at 18 °C – 24° C (30–70% room humidity) and housed in a PC2 facility with ad libitum access to food and water. Mice used were wildtype (WT) mice (C57BL/6 strain), Tau knockout (KO) mice[79] and TALEN gene-edited Tau-mEos2 mice[55], both generated on a C57BL/6 background. Primary hippocampal neurons were obtained from WT, and Tau KO and Tau TALEN at embryonic day (E) 16. Isolated hippocampi were prepared as previously described[9]. Briefly, for live-cell super-resolution microscopy, 100,000 neurons were seeded onto poly-L-lysine-coated (1 mg/ml) 35 mm glass-bottom dishes (In Vitro Scientific) in Neurobasal medium (Gibco) supplemented with 5% fetal bovine serum (Hyclone), 2 mM Glutamax (Gibco) and 50 U/mL penicillin/streptomycin (Invitrogen). The medium was changed to serum-free Neurobasal medium supplemented with 2% B27 (Gibco) 3 h post-seeding, and half the medium was changed every week.

### Heterologous cell cultures
HEK-293T cells (293 T/17 [HEK 293 T/17] (ATCC® CRL11268™)) were maintained in DMEM medium (GIBCO-Thermo Fisher Scientific) supplemented with 10% FBS (GIBCO-Thermo Fisher Scientific), 1x Gluta-MAX (GIBCO-Thermo Fisher Scientific) and 100 U/ml penicillin-100 µg/ml streptomycin (Sigma-Aldrich-Merck). Cells were seeded onto poly-L-lysine-coated (0.1 mg/ml) 35 mm glass-bottom dishes and transfected using the Lipofectamine™ LTX reagent according to the manufacturer's instructions (Invitrogen-Thermo Fisher Scientific).

### Constructs
VAMP2-pHluorin plasmid was kindly provided by J. Rothman (Yale University, New Haven, CT, USA)[80]. mEos2-N1 was a gift from Michael Davidson & Loren Looger (Addgene plasmid # 54662; http://n2t.net/addgene:54662; RRID:Addgene_54662)[81]. mCardinal-N1 plasmid was a gift from Michael Davidson (Addgene #54590; http://n2t.net/addgene:54590)[82]. Tau-mEos2 (human Tau with carboxy-terminal mEos2 tag)

and GFP-Tau were kindly provided by J. Götz (The University of Queensland, Brisbane, QLD, Australia)[83]. vGLUT1-pHluorin and vGLUT-mEos2 were kindly provided by D. Perrais and E. Herzog (University of Bordeaux, Bordeaux, France). The Tau mutant Tau[I277P / I308P] was generated by replacing two isoleucines by prolines at the 277 (I277P) and the 308 (I308P) positions of the Tau gene of the Tau-mEos2 plasmid. Mutations were generated using the Quick-change Multi-site-Directed mutagenesis kit (Agilent). Primers were designed using PrimerX (https://www.bioinformatics.org/primerx) (Tau I277P forward primer 5′-GGCGGGAAGGTGCAGCCAATTAATAAGAAGC-3′; Tau I308P forward primer 5′-GCGGCAGTGTGCAACCAGTCTACAAACCAG-3′). Tau-GFP and Tau[I227P, /I308P]-GFP were created by restriction digestion of Tau ORF from Tau-mEos2 or Tau[I227P/I308P]-mEos2 with SalI and AgeI and subsequent T4 Ligation (New England Biolabs) in the pEGFP-N1 plasmid. The Tau construct ΔTau74-mEos2 was generated by subcloning ΔTau74 from ΔTau74-GFP[51,54], by PCR amplification and recombination using InFusion (Clontech-Takara Bio) between XhoI/EcoRI of the linearized pmEos2-N1 plasmid. ΔTau74 was amplified (using the following primers forward (5′-GACTCAGATCTCGAGGCCACCATGGCTGAG CCCCGC-3′; reverse 5′- CGTCGACTGCAGAATTCGAAGATTC TTCAGGTCTGGCATGGGCAC-3′) and positive clones were confirmed by Sanger sequencing at the Australian Genome Research Facility (AGRF, Brisbane, Australia).

## Transfections

Primary neurons were transfected at DIV 12–15 using the Lipofectamine 2000 (Invitrogen Thermo Fisher Scientific) reagent, as previously described[9]. HEK-293T cells were transfected with lipofectamine 3000 (Thermo Fisher Scientific) according to the manufacturer's instructions. Transfected cells were incubated for 24–48 h prior to imaging.

## Super-resolution microscopy with oblique illumination

Primary transfected neurons were used for super-resolution microscopy 2–7 days post-transfection. The following constructs were used for transfection: VAMP2-pHluorin, vGLUT1-mEos2, mEos2-N1, mCardinal-N1 and Tau-mEos2. For live-cell super-resolution microscopy with oblique illumination, transfected neurons were visualized at 37 °C on a Roper Scientific Ring-TIRF microscope equipped with an iLas2 double laser illuminator (Roper Scientific, FL, USA), a Nikon CFI Apo TIRF 100/1.49 N.A. oil-immersion objective (Nikon Instrument, NY, USA) and a Perfect Focus System (Nikon Instruments, NY, USA). Imaging was performed using two Evolve512 delta EMCCD cameras (Photometrics, AZ, USA) mounted on a TwinCam LS Image Splitter (Cairn Research, UK), a quadruple beam splitter (ZT405/488/561/ 647rpc, Chroma Technology, VT, USA) and a QUAD emission filter (ZET405/488/561/640 m, Chroma Technology, VT, USA).

## Photoactivated localization microscopy (sptPALM), subdiffractional tracking of internalized molecules (sdTIM) and dual-color super-resolution imaging

Presynaptic compartments (axon and presynapses) from neurons positive for VAMP2-pHluorin or vGLUT1-mEos2 were selected using the green channel. During imaging, neurons were incubated in low K+ resting imaging buffer (145 mM NaCl, 5.6 mM KCl, 2.2 mM CaCl₂, 0.5 mM MgCl₂, 5.6 mM D-glucose, 0.5 mM ascorbic acid, 0.1% BSA, 15 mM HEPES, pH 7.4). Neuronal stimulation was performed by incubating the neurons with high K+ stimulating imaging buffer (95 mM NaCl, 56 mM KCl, 2.2 mM CaCl₂, 0.5 mM MgCl₂, 5.6 mM D-glucose, 0.5 mM ascorbic acid, 0.1% BSA, 15 mM HEPES, pH 7.4). Time-lapse movies (16,000 frames) were acquired at 50 Hz using Metamorph software (version 7.7.8, Molecular Devices, CA, USA). For sptPALM, a 405-nm laser was used to photo-convert vGLUT1-mEos2 or Tau-mEos2, and a 561-nm laser was used simultaneously for excitation and bleaching of the resulting photo-converted single molecules. To isolate the mEos2 signal from autofluorescence and background signals, a

double-beam splitter (LF488/561-A-000, Semrock, NY, USA) and a double-band emitter (FF01-523/610-25, Semrock, NY, USA) were used. To spatially distinguish and temporally separate the stochastically activated molecules during acquisition, the 405-nm laser was used between 1.5% and 5% of the initial laser power (100 mW Vortran Laser Technology), and the 561-nm laser was used at 70% of the initial laser power (150 mW Cobolt Jive). sdTIM was performed to track the mobility of VAMP2-pHluorin-containing single recycling vesicles as described previously[10]. Briefly, anti-GFP nanobodies tagged with Atto-647 (Synaptic Systems) were diluted in high K+ stimulation imaging buffer at a concentration of 100 pM Atto-647 nanobodies. Neurons were stimulated for 5 min, the stimulation buffer was then removed and neurons were washed 3–5 times in imaging buffer. Cells were incubated for 10 min in imaging buffer (37 °C and 5% CO₂) before being imaged. sptPALM and sdTIM were simultaneously combined to provide dual-color super-resolution imaging of Tau-mEos2 and single recycling vesicles. Dual-color imaging was performed as described previously[84]. Two EMCCD cameras were mounted on a TwinCam LS Image Splitter, allowing fine xy-alignment of both the transmitted and reflected ports. TetraSpeckTM microspheres (Invitrogen-Thermo Fisher Scientific, 0.1 μm) diluted in imaging buffer were used to calibrate the correct alignment of the cameras.

## Single-particle trajectory analysis

The localization and tracking of single molecules were performed as previously described[85]. Briefly, single-molecule localizations were detected using a wavelet-based segmentation, and trajectories were computed using a simulated annealing-based tracking algorithm[86] with PALM-Tracer (version 2.1.0.28228), a custom-written software package tool that operates as a plugin of Metamorph (Molecular Devices)[85,87]. Regions of interest (ROIs) were drawn around nerve terminals, defined as hotspots of increased pHluorin fluorescence. Trajectories that lasted at least eight frames were reconstructed and the mean square displacement (MSD) was computed for each trajectory. The MSD was fitted by the equation $MSD(t) = a + 4Dt$ (where $D$ = diffusion coefficient, $a$ is y intercept and $t$ is time). Mean-square displacement (μm²) is calculated and plotted over a 0.2 s time frame. The frequency distribution of diffusion coefficient (Log10 of μm² s⁻¹) is quantified. Trajectories were assigned as immobile when their $Log10[D] \leq -1.6$ μm² s⁻¹ [86–88]. We computed the % of the mobile fraction from the distribution of the diffusion coefficient histograms for statistical comparisons. Single-particle trajectory data generated in this study are available for download from the publicly accessible institutional data repository of The University of Queensland (UQ eSpace) 10.48610/a9c9def.

## FRAP imaging and analysis

HEK-293T cells expressing the Tau-GFP were imaged using a LSM 710 Inverted point-scanning laser confocal microscope with a 63x / 1.4 NA oil objective, equipped with spectral detection and high-sensitivity BiG (GaAsP) detectors, and CO2 and temperature controllers. Cells were excited with a 488 nm Argon laser. The FRAP protocol was divided into three sections: initially, cells were imaged 5 times every 1.16 s (pre-bleach). Then, a small selected circular region (2.108 μm ∅) containing Tau-GFP was photobleached, using a 488 nm Argon laser at 80% (dwell time: 35 ms, 100 repeats). After the bleaching, the same region was scanned 50 times every 1.16 s (postbleach). FIJI-ImageJ software was used to measure intensities from the acquisitions. For each time point, the whole cell fluorescent intensity ($I_{whole}$), the background fluorescent intensity ($I_{background}$) and the photobleached fluorescence recovery ($I_{FRAP}$) integrated density were measured. Recovery intensities were first normalized to the average prebleach fluorescence intensity. Whole-cell and recovery intensities were corrected by subtracting background intensities. Finally, whole-cell intensities were used to correct the recovery intensities, needed to correct for photobleaching due to the acquisition protocol. The formula to calculate

FRAP curves is the following:

$$I_{FRAPnorm}(t) = \frac{I_{whole-prebleach}}{I_{whole}(t) - I_{background}(t)} \times \frac{I_{FRAP}(t) - I_{background}(t)}{I_{FRAP-prebleach}}$$

Once normalized intensities were calculated at each time point, averages and SEM values were plotted using GraphPadPrism and the resulting FRAP curve was fitted using nonlinear regression to a one-phase decay curve.

## Immunofluorescence

Primary neurons were imaged in either imaging buffer as a control or in stimulation buffer (high K$^+$) for 3 min to initiate synaptic activity. HEK-293T cells and neurons were fixed in 4% paraformaldehyde in PBS for 30 min at room temperature (RT). They were then washed in PBS and permeabilized with 0.1% Triton x100 (Sigma-Aldrich-Merck) for 4 min. After permeabilization, non-specific binding sites were incubated in blocking solution (5% horse serum, 1% BSA in PBS) for 30 min at RT. Primary antibodies were diluted in primary antibody solution (1% BSA in PBS) and incubated for 1 h at RT. The following primary antibodies were used: anti-MAP2 guinea pig polyclonal antibody (1:2500, Synaptic Systems, #188004), anti-VAMP2 mouse monoclonal antibody (1:500, Synaptic Systems, #104211), anti-Rab5 rabbit monoclonal antibody (1:1000, Abcam, #ab218624), anti-KDEL rabbit monoclonal antibody (1:200, Abcam, #ab176333), anti-Lamp1 mouse monoclonal antibody (1:100, Abcam, #ab25630), anti-GFP chicken polyclonal antibody (1:1000, MERK, #AB16901), anti-Tau mouse monoclonal antibody T46 (1:100, ThermoFisher, #13–6400), and anti-Tau clone Tau-5 mouse monoclonal antibody (1:1000, Sigma-Aldrich -Merck, #MABN162). The Tau-5 antibody recognizes the tau molecule's mid-domain (amino acids 210–241) regardless of its phosphorylation state[89–91]. Thus, it indicates the total amount of tau (phosphorylated and non-phosphorylated) within the system. Secondary antibodies were diluted in secondary antibody solution (5% horse serum in PBS), and incubated for 30 min at room temperature. The following secondary antibodies were used: Alexa Fluor® 488 conjugated anti-mouse IgG (#A-32723), Alexa Fluor® 488 conjugated anti-chicken IgG (1:500, Thermo Fisher Scientific; #A-11039), Alexa Fluor® 546 conjugated anti-guinea pig IgG (#A-11074), Alexa Fluor® 555 conjugated anti-rabbit IgG (#A-32732), Alexa Fluor® 555 conjugated anti-mouse IgG (#A-32727), Alexa Fluor® 546 conjugated anti-chicken IgG (#A-11040), Alexa Fluor® 647 conjugated anti-rabbit IgG (#A-31573) and Alexa Fluor® 647 conjugated anti-guinea pig IgG (#A-21450) (all diluted 1:500, Thermo Fisher Scientific). Finally, cells and neurons were washed and mounted in Vectashield® Plus antifade mounting medium (Vector Laboratories).

## Confocal imaging for colocalization analysis

Confocal images on Supplementary Fig. S8 and S10 were acquired with a Zeiss Plan Apochromat 63x/1.4 NA oil-immersion objective on a confocal/two-photon laser-scanning microscope (LSM 710; Carl Zeiss) built around an Axio Observer Z1 body (Carl Zeiss), equipped with two internal gallium arsenide phosphide (GaAsP) photomultiplier tubes (PMTs) and three normal PMTs for epifluorescence detection. The system was controlled by the Zeiss Zen Black software. After acquisition, images were further processed and analyzed with FIJI-ImageJ[92]. Confocal images on Supplementary Fig. S12 were acquired with a spinning-disk confocal system consisting of an Axio Observer Z1 equipped with a CSU-W1 spinning-disk head, ORCA-Flash4.0 v2 sCMOS camera, 20 × 0.8 NA PlanApo and 40 × 1.2 NA C-Apo objectives. Image acquisition was performed using SlideBook 6.0. Sections of each slide were acquired using a Z-stack with a step of 130 nm. The exposure time for each channel was kept constant across all imaging sessions. Images were deconvolved on Huygens deconvolution software. The maximum intensity projection of each Z-stack was analyzed in FIJI-Image J[92]. Masks regions of interest (ROI) were created using anti-VAMP2 as a synaptic marker, and the Tau-5 mean grey intensity was measured in those regions.

## Hidden Markov modelling

We used the the variational Bayes SPT (vbSPT) analysis to infer the number of hidden diffusive states from VAMP2-pHluorin trajectories as previously described[37]. We allowed a maximum of three hidden states, with each state representing immobile, confined and highly mobile diffusive states[38]. This analysis includes 131 presynapses from sdTIM experiments and 35 presynapses from sptPALM experiments, all containing over 100 trajectories per presynapse. All the analysis were performed using MATLAB (MathWorks, Inc.).

## Nanoscale spatiotemporal indexing clustering of trajectory segments (segNASTIC)

Cluster analysis was performed using segNASTIC software (v20220221) as previously described[49]. For this analysis, a maximum of 20 trajectories per m$^2$ were randomly chosen by the software to represent the total number of detections in each acquisition. Trajectory segments in the same ROI had to overlap four times more than the average overlap threshold and appear within ±20 s of each other to be considered clusters. Metrics generated by the segNASTIC software includes membership (traj/cluster), cluster lifetime (s), cluster average MSD (m$^2$), cluster area (m$^2$), cluster radius (μm), density in clusters (traj/m$^2$), rate (traj/s), percentage clustered trajectories (%) and cluster density (clusters/m$^2$). If an ROI was found to be an outlier in five or more of these metrics it was removed from the analysis. To compare cluster metrics from multiple acquisitions and/or conditions, the NASTIC wrangler software was used to compile the data for statistical analysis[49].

## Pharmacological control of phosphorylation and liquid-liquid phase separation

Pharmacological modification of protein phosphorylation was performed by incubating neurons with the pan protein phosphatase inhibitor okadaic acid (OkAc)[70] (Sigma-Aldrich-Merck), or with the non-selective protein kinase inhibitor staurosporine (Staur)[93] (Sigma-Aldrich-Merck). Both reagents were incubated at 1 μM for 15 min at 37 °C. Liquid-liquid phase separation (LLPS) of Tau protein was dissipated with the aliphatic alcohol 1,6-hexanediol (1,6-HD)[94] (Sigma-Aldrich-Merck). 1,6-HD (2.5%) was prepared in imaging media (phenol-free neurobasal). Neurons were incubated with 1,6-HD media for 5 min and thereafter imaged in the continuous presence of 1,6-HD for up to 15 min.

## Protein digestion and phosphopeptide enrichment

We generated the protein samples by stimulating mouse hippocampal neurons through depolarization with high K$^+$ buffer before lysis, protein precipitation and tryptic digestion. Lysis buffer (2% SDS, 50 mM HEPES/NaOH, pH 7.4, 2 mM EDTA, 2 mM EGTA, Roche COMPLETE protease inhibitor, Roche PHOSSTOP phosphatase inhibitor) was used to obtain four lyophilized lysates each of low K$^+$ and high K$^+$ treated hippocampal neurons, which were then dissolved in 200 μL of MilliQ water (18 MΩcm). Tris(2-carboxyethyl)phosphine (TCEP) was added at 5 mM and the samples were incubated at 80 °C for 10 min. Samples were cooled to 23 °C and iodoacetamide was added at 20 mM and kept in the dark for 30 min. The samples were precipitated using the chloroform-methanol method[95] and dried at 40 °C for 1 h. Samples were resuspended in 20 μL of 7.8 M urea buffered with 100 mM HEPES pH 8.0. To each sample was added 3 μg of LysC (Fujifilm Wako Pure Chemical Corporation) in 3 μL. Samples were incubated at 25 °C for 8 h with shaking. Samples were diluted by the addition of 140 μL of 100 mM HEPES pH 8.0 and 5 μg of trypsin (TrypZean, Sigma) was added in 5 μL. Samples were digested for 6 h at 25 °C with shaking. Peptide concentration was determined using UV absorbance at

280 nm light (Implen Nanophotometer, Labgear). Phosphopeptides were enriched from the tryptic peptides and derivatized with TMT10plex reagents to create an 8-plex sample. For each sample, 100 µg was labelled with TMT10/11plex reagent according to the manufacturer's instructions (Thermo Fisher Scientific). The following labelling scheme used. Low $K^+$ experiment 1, 127 C; high $K^+$ experiment 1, 128 N, low $K^+$ experiment 2, 128 C; high $K^+$ experiment 2, 129 N; low $K^+$ experiment 3, 129 C; high $K^+$ experiment 3, 130 N; low $K^+$ experiment 4, 130 C; high $K^+$ experiment 4, 131 N. Samples were checked for derivatization efficiency (>98%) and then, quenched, combined, acidified and desalted with a Sep-Pak tC18 3cc Vac cartridge (200 mg sorbent, Waters). Phosphopeptides were enriched and fractionated using a previously described "TiSH" method, resulting in 17 phosphopeptide fractions separated by hydrophilic ion chromatography (HILIC). The HILIC was performed using a Dionex UltiMate 3000 HPLC system (Thermo Fisher Scientific) with a TSKgel Amide-80 1 mm inside diameter × 250 mm long column (Tosoh Bioscience). Fractions were collected using a Probot (LC Packings).

## Mass spectrometry
The 17 fractions of the 8-plex sample were each analyzed by LC-MS/MS using a 2 h instrument method. Each fraction was loaded onto an in-house packed 300 × 0.075 mm C18 column (ReproSil Pur C18 AQ 1.9 µm particles, Dr Maisch, Germany) by a Dionex UltiMate 3000 RSLC nano system. The column was heated at 50 °C using a column oven (PRSO-V1, Sonation lab solutions, Germany) integrated with the nano flex ion source of the Q Exactive Plus hybrid quadrupole-orbitrap mass spectrometer (Thermo Fisher Scientific). The sample was loading was for 25 min in 99% phase A (0.1% formic acid in water) and 1% phase B (0.1% formic acid, 90% acetonitrile, 9.9% water) at 300 nL/min. The gradient was at 250 nL/min from 5% phase B to 25% phase B in 74 min, then to 35% phase B in 8 min, then to 99% phase B in 1 min, held at 1% phase B for 2 min, then to 1% phase B in 1 min and held at 1% phase B for 8 min. The spray voltage was 2.3 kV. The capillary temperature was 250 °C. The S-lens radio frequency level was 50. Peptides were selected for MS/MS using data-dependent acquisition from MS scans from m/z 375 to 1500 at a resolution of 70,000 with an automatic gain control target of 1,000,000 and a maximum scan time of 100 ms. The top 12 most intense peptides were selected for an MS/MS scan using an isolation window of m/z 1.2 with a fixed first mass at m/z 120 at a resolution of 35,000 with an automatic gain control target of 200,000 and maximum scan time of 115 ms. The normalized collision energy was 34. Single-charged ions and those with charges >8 were excluded. Dynamic exclusion was applied for 35 s.

## Mass spectrometry data processing
LC-MS/MS analysis of the 8-plex identified 20,395 phosphopeptides. Raw LC-MS/MS data files were processed with MaxQuant v1.6.7.0[96] with the following settings. The FASTA file was the *Mus musculus* reference proteome with canonical and isoform sequences downloaded from UniProt on 21st February 2020. For comparisons with human Tau, the FASTA file was the *Homo sapiens* 2N4R (Tau-F, Tau-4) sequence downloaded from UniProt on 12th October 2023. The in-built contaminants FASTA file was included. Digestion was set as specific for trypsin with cleavage at Arg-Pro allowed and up to three missed cleavages allowed. Carbamidomethyl modification of Cys was a fixed modification. Phosphorylation (STY), N-terminal acetylation, oxidation (M) and deamidation (NQ) were variable. TMT reporter ion tolerance was 0.003 Da and precursor intensity fraction was required to be greater than 0.6. The minimum peptide length was 6 and the maximum peptide mass was 6000 Da. Second peptides and dependent peptides were enabled. The protein identification and peptide spectrum matching false discovery rates were at the default 1% (compared to the reversed database) and the minimum score for modified peptides was at the default 40. All other settings were default. The

mass spectrometry proteomics data and MaxQuant output have been deposited to the ProteomeXchange Consortium via the PRIDE[97] partner repository with the dataset identifier PXD020232 and 10.6019/PXD020232.

The 'evidence.txt' MaxQuant output file was used as input to a process of data refinement described previously[98]. Briefly the following processes were implemented. Reverse database hits and contaminants were removed. All phosphosites were remapped to the same mouse proteome used for MaxQuant processing and where multiple UniProt protein accessions matched a peptide, a single accession with the highest evidence for existence was assigned. Phosphopeptides with a localization score less than 0.75 were filtered out and the median intensities of redundant phosphopeptides were assigned. A non-zero intensity value was required for each of the TMT reporter ion channels used. Intensities were $log_2$ transformed, quantile normalized and batch corrected using surrogate variable analysis. High and low $K^+$ samples were highly correlated within each group. Low $K^+$ was compared to high $K^+$ using limma statistics. P-values derived from a moderated t-test were adjusted for false discovery and anything less than 5% was considered significant.

Up-regulated phosphorylation of presynaptic phosphoproteins was compared to the disordered protein sequence using the following process. First, phosphopeptides that were not significantly regulated in the high $K^+$ condition versus low $K^+$ were excluded (i.e., resulting in a positive value indicating up-regulation of phosphorylation). Synaptic proteins were defined using Synaptic Gene Ontologies (SynGO[45]) release 1.0 (1st April 2019) and proteins not recorded in SynGO as synaptic proteins were excluded. An exception was made for tau, which is currently not annotated in SynGO. For each presynaptic protein, the phosphopeptide with the maximum positive $log_2$ intensity differential between high $K^+$ stimulation and low $K^+$ resting was identified. Thus, each presynaptic protein was assigned a maximum up-regulated phosphorylation value, expressed as a $log_2$ intensity differential. The same process was used to obtain a maximum $log_2$ intensity differential for the presynaptic proteins stimulated with 76 mM KCl (high $K^+$) compared to non-stimulated samples incubated with 4.7 mM KCl (low $K^+$), in the work by Engholm-Keller et al.[44]. Upregulated phosphorylation was then compared to the predicted or, when available, the experimentally determined percentage of intrinsically disordered sequence for each presynaptic protein, as recorded in the MobiDB and DisProt databases. Values for the predicted percentage of disordered sequence content were obtained by downloading (6th March 2020) the *Mus musculus* and *Rattus norvegicus* data from MobiDB, calculated using the MobiDB lite algorithm[99]. UniProt protein accessions were used to match the disorder data to phosphoproteomics data and were then reduced to a list of non-redundant gene names by discarding the smallest $log_2$ intensity differential. If a phosphopeptide matched to more than one gene name, then the first gene name reported by our data refinement method was used. Where available, experientially determined and manually curated values for percentage disorder downloaded from DisProt[100] were used instead of MobiDB data. The September 2019 release of mammalian DisProt data was used, which contained 717 entries. DisProt data for *Mus musculus* and *Rattus norvegicus* was used when available, but if not, the *Homo sapiens* data for the same gene name was used, rather than the MobiDB predictions. For phosphosite alignments between the 'High vs Low' *Mus musculus* and *Rattus norvegicus* 'Engholm-Keller' stimulated data, each protein identified in the High vs Low experiment was fully aligned to its homologous *Rattus norvegicus* protein using the BLOSUM62 algorithm.

## Statistical analysis
Results were analyzed statistically using GraphPadPrism software (GraphPad Software, Inc). The D'Agostino and Pearson test was used to test for normality. The unpaired two-tailed Student's t-test was used

for comparison of two groups when the data were normally distributed, and the non-parametric Mann Whitney *U* test was used when the data were not normally distributed. For datasets comparing two (normally distributed) paired groups, the Student's paired *t*-test was performed. For datasets comparing more than two groups, we performed the Brown-Forsythe and Welch ANOVA test followed by Dunnett t3 *post hoc* multiple comparisons test, or the Kruskal-Wallis' test followed by Dunn's *post hoc* test corrections for multiple comparisons. Statistical comparisons were performed on a per-cell or per neuronal region (axon or presynapse). The neurons analyzed for each experiment were derived from pooling ≥ 5 dissected embryos. In experiments performed on HEK-293T cells, the statistical comparisons were performed on a per-experiment or a per-cell basis, as specified in the figure legends. All data points lying over two standard deviations from or above the mean were considered outliers and removed from the dataset. This exclusion criteria was pre-established. These outliers were identified using the Outlier Wrangler[49] custom-made Python script. In the mass spectrometry and proteomic experiments, single-charged ions and those with charges >8 were excluded, and phosphopeptides that were not significantly regulated in the high $K^+$ condition versus low $K^+$ were excluded. Neurons were collected from at least two independent experiments. Values are represented as the mean ± SEM. The tests used are indicated in the respective figure legends. A *p*-value below 0.05 was considered significant and the numerical value was indicated in the graphs.

### Reporting summary

Further information on research design is available in the Nature Portfolio Reporting Summary linked to this article.

## Data availability

Source data are provided in this paper. The accession codes of the datasets deposited to ProteomeXchange Consortium via the PRIDE are PXD020232 and 10.6019/PXD020232. The MobiDB (https://mobidb.bio.unipd.it/) and DisProt (https://disprot.org/) databases were used to experimentally determine the percentage of intrinsically disordered sequence for each presynaptic protein. The *Mus musculus* and *Rattus norvegicus* data from MobiDB, and the *Mus musculus*, *Rattus norvegicus* and *Homo sapiens* data from DisProt were downloaded to obtain the values for the predicted percentage of disordered sequence content. The stimulated data from the manuscript Engholm-Keller et al.[44] was used to align each phosphorylated protein to its homologous *Rattus norvegicus* in the experiment of neuronal stimulation with high $K^+$. Single-particle trajectory data generated in this study are available for download from the publicly accessible institutional data repository of The University of Queensland (UQ eSpace) 10.48610/a9c9def. The dataset generated and analysed during the current study is available in the Figshare repository (https://doi.org/10.6084/m9.figshare.23895768). Source data are provided with this paper.

## Code availability

Computer codes to analyze the data have been described in the work by Wallis et al.[49]. All Python codes used are available at the GitHub repository https://github.com/tristanwallis/smlm_clustering. The *NASTIC* suite v1.0.0 is available at https://doi.org/10.5281/zenodo.7847776 under a Creative Commons CC BY 4.0 licence: you are free to use and modify the code on the proviso that you make any changes freely available, acknowledge the original authors in derivative works and do not release said works under a more restrictive licence.

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

## Acknowledgements

The super-resolution imaging was carried out at the Queensland Brain Institute's (QBI's) Advanced Microimaging and Analysis Facility. We thank Dr. Rumelo Amor for expert technical help with microscopy, and Dr. Alex McCann for critically reading and editing the manuscript. We acknowledge Corey Butler and Adel Kechkar for their contributions in the development of PALM-Tracer. The TALEN Tau-mEos2 animals were kindly provided by Prof. Jürgen Götz. This work was supported by grants from the Australian Research Council Discovery Project grant 170100125 (F.A.M.), the National Health and Medical Research Council (NHMRC) grant 1139316 (F.A.M.), the Clem and Jones Foundation (R.M.M.), the National Institute On Aging of the National Institutes of Health (NIH) grant R21AG080435 (F.A.M.), the UQ Research Stimulus Allocation 2 fellowship (R.M.M.), the NHMRC Boosting Dementia Research Initiative (R.M.M.), the UQ Early Career Research Grant 2057309 (M.J.), the Australian Research Council Discovery Early Career Researcher Award DE190100565 (M.J.), the Research Training Program (RTP) Scholarship and a QBI top-up scholarship (C.S.), and the Australian Research Council Linkage Infrastructure, Equipment, and Facilities grant LE130100078 (F.A.M., QBI Advanced Microimaging and Analysis Facility). Graphic schemes in Fig. 7 were created with Biorender.com.

## Author contributions

R.M.M. and F.A.M. were responsible for the conceptualization and planned and oversaw all aspects of the study; R.M.M., M.J., T.P.W.,

M.E.G., and A.J.W. were involved in setting up all methodological procedures required during this investigation; R.M.M., S.F.L., M.M., M.J., J.R.W., A.J.W. and C.S. performed and analyzed most of the experiments; R.M.M., S.F.L., M.J., M.M. and C.S. performed all super-resolution imaging; R.M.M., S.F.L., M.J., M.M. and T.P.W. analyzed the super-resolution data; R.S.G. performed all the cloning and muta-genesis; R.M.M., J.R.W., A.J.W. performed the phosphoproteomic experiments; M.E.G. and A.J.W. analyzed the prosphoproteomic results; F.A.M. and M.J. were involved in funding acquisition; F.A.M. led the project administration; R.M.M., F.A.M., M.E.G. and M.J. supervised the research; R.M.M., S.F.L., M.E.G. and F.A.M. wrote the original draft of the paper; R.M.M., S.F.L., T.P.W. and F.A.M. wrote, reviewed and edited the final version of the paper with input and substantial revi-sions from all authors.

## Competing interests

The authors declare no competing interests.
