## [Peer Review File · Nature Communications]

Tau forms synaptic nano-biomolecular condensates controlling the dynamic clustering of recycling synaptic vesiclesEditorial Note: Parts of this Peer Review File have been redacted as indicated due to the presence of confidential information.

Reviewers' comments:

Reviewer #1 (Remarks to the Author):

Martínez-Mármol et al. report in their manuscript that Tau forms presynaptic nanoclusters and regulates the dynamic of the recycling synaptic vesicle pool. After identifying Tau as a protein that gets phosphorylated upon hyperexcitation and being a phase separating IDP in neurons, they reason that Tau might be a regulator of SV dynamics, in analogy to Synpsin1. Using single molecule particle tracking techniques, the authors compare the trajectories of Tau-mEOS2 in presynapses with that of axonal Tau. They find that Tau forms 'nanoclusters' in presynapses and seems to regulate the decrease the diffusion dynamics of recycling synaptic vesicles. The authors then argue that the Tau clusters are biomolecular condensates based on their sensitivity to hexanediol.

This is a nice story showing the clustering of Tau in presynapses and the effect on vesicle dynamics, however, besides these two aspects most reported findings have previously been shown. The story lacks novelty and conclusive experiments that support the conclusion of biomolecular condensation of Tau in the presynapse.

Major concerns:

- The claimed formation of biomolecular condensates of Tau in presynapses - which by definition in the Tau and phase separation communities should have liquid-like properties - is not supported by the data, and the title and conclusion is therefore misleading. The HEK cell images showing large dots of GFP-Tau do not support the conclusion of condensation either bc they look rather lysosomal. Those are likely not Tau condensates. Using sensitivity to hexanediol as the only sign of being a liquid condensates is not accepted because HD changes and dissolves a lot of things in cells!

- The introduction has too little information about Tau in general and presynaptic Tau, and important relevant landmark references are missing! e.g. general Tau reference and Tau on MT reference has to be given, reference(s) to Roland Brandt's lab who showed the binding activity of Tau to MTS by single particle tracking in 2014, and more recent work of his lab on Tau tracking in stress granules. These are important papers that report already many things presented as findings in this manuscript.

- Activity dependent phosphorylation of Tau and its status of an IDP are known since a long time. The data presented here do not present any novel aspect of this, they do not show an increase in synaptic but in total Tau phosphorylation, and even lack the information about what kind of phosphorylation is found on Tau upon K⁺ induced neuronal activation. Neither is the increase of phosphorylation shown by IHC or Biochemistry.

Reviewer #2 (Remarks to the Author):

Martinez-Marmol et al investigate the role of the Tau protein for synaptic vesicle organization and distribution in the presynaptic compartment. They make use of an exploratory phosphoproteomics approach, identifying Tau as a potential key player, and then use single-molecule super-resolution microscopy and biochemical assays to further characterize the role of Tau in activity-dependent regulation of synaptic vesicle nanoscale organization and function. They propose that liquid-liquid phase separation of Tau could play a central role in these processes.

The study is well-designed, the experiments are conducted thoroughly and the manuscript is well written. The authors address an important topic in neurobiology, how synaptic vesicles are organized at the presynapse. I thus recommend publication in Nature Communications after minor revisions, as outlined below:

- Expand or provide a more detailed analysis of the phosphoproteomics results: The phosphoproteomics data are presented in a protein-centric manner: A single phosphopeptide with the highest fold change per protein was used to show differential protein phosphorylation upon activity stimulation. However, proteins could also display a decrease in phosphorylation upon activity stimulation - this would be completely lost in this analysis. Also, multiple, reciprocal phosphorylation events on the same protein would be missed. For simplicity and to get an overview, the analysis of the authors is ok but we kindly ask the authors to include a complete overview of activity-dependent phosphorylation and dephosphorylation events on a phosphosite level in the Supplementary data (e.g. in a figure and a table). In addition, an overview of the phosphoproteomics data prior to filtering with SynGO in a supplementary figure would be appreciated.

- Show controls for or comment on Tau liquid-liquid phase separation experiments in HEK cells: The authors should comment on their experimental design when overexpressing Tau in HEK cells with a GFP-tag, and include a separate experiment with a similar-sized control protein at similar expression levels not showing phase separation in the same conditions (or provide a similar reference). Such a control would significantly strengthen the manuscript.

- Comment in more detail on the effects of phosphatase and kinase inhibitors in the Tau knockout: While the absence of OkAc-induced increase in recycling vesicle mobility in a Tau knockout is in line with the authors hypothesis, staurosporine treatment still lead to a decrease in recycling vesicle mobility. These observations could be discussed in the Discussion section, as they indicate additional components responsible for SV mobility which are dependent on phosphorylation but not on Tau.

In the same context, the authors could comment on the known targets of the kinase and phosphatase inhibitors used in this study.

- Provide more details for experimental conditions and data availability: In the sample preparation section, we found no mention of phosphatase inhibitors being used during sample preparation for the phosphoproteomics samples. Is this correct? Please also include the K concentrations of the "high/low conditions" earlier in the method section and mention them at least once in the results section (at

present, the conditions are only mentioned once at the end of the method section). Please also include access to the proteomics data uploaded to PRIDE, as required by the Nature Communications guidelines for authors.

Reviewer #3 (Remarks to the Author):

The work by Martinez-Marmol et al. uses relatively novel and highly sophisticated approaches for super-resolution single molecule tracking applied to different pools of synaptic vesicles (total using sptPALM or recycling sdTIM) as well as tracking tau in the axon and synapses of cultured neurons. These imaging approaches have facilitated some novel and impactful findings (noted below) that would advance the field. However, there are some issues that should be addressed. In particular, the entire focus on LLPS is not well supported and additional studies to further substantiate some aspects of the work and better discussion of the results is needed. Specific comments are provided below.

The work showing tau's role in regulating the mobility of the recycling pool of SVs is a novel finding and of high importance to better understanding the multi-functional roles of tau protein in neurons. Further, the differences in the properties of axonal tau clusters and synaptic tau clusters is an interesting novel finding.

The mass spectrometry work assessing the synaptic phosphoproteome with or without potassium stimulation is novel and provides compelling data on proteins modified in an activity-dependent fashion. However, there are no data convincingly demonstrating that the phosphorylation changes in tau are due specifically to synaptic tau as opposed to tau in any of the other cellular compartments (e.g. dendrites, cell bodies axons, etc.). The phosphorylation status of tau (e.g. sites and localization) was not assessed in the synaptic- and axon-focused experiments, which is a missed opportunity to provide potentially insightful data on specific tau phospho-changes associated with the different observations of tau behavior. Further, localization of pTau changes in the presynaptic space would strengthen this point and the involvement of synaptic tau phosphorylation in modulating SV mobility.

Generally, the data supporting the nanoclusters represent LLPS structures is not strong or well-supported in neurons, which is a central point of the manuscript. An area simply containing more protein than the surrounding area does not in and of itself constitute a LLPS structure. The nanocluster designation can stand alone without being described as LLPS structures, but if LLPS remains it must be further substantiated. The HEK cell work does not help to support neuronal LLPS and seems somewhat disconnected from the rest of the work presented in neurons.

How were axons and synapses operationally defined? It is unclear how the authors confirmed (i.e. what metrics and/or markers were used) they were performing the tracking analyses in axons and bona fide synapses. Clarification and/or confirmatory studies are necessary.

There is no consideration/discussion of the focus on en passant synaptic structures versus terminal

synaptic structures in the cultured neuron and whether differences may exist in tau's behavior/role in each. It is also not clear if the synapses being analyzed are functionally connected synapse and whether this could alter the observations made in these studies.

The introduction lacks introductory components on tau (a central component of the manuscript) and the discussion is extremely shallow.

There are a few minor misstatements: The epitope for Tau5 is incorrect (updated epitope - PMID: 17499212; original epitope - PMID: 8955115; original paper - PMID: 7479786). Tau is not exclusive to the nervous system.

Point-by-point reply to the reviewers' comments and criticism

We are grateful to the reviewers for their thoughtful and constructive evaluation of our manuscript. We agreed with the suggested changes and have revised the manuscript accordingly. Below is a point-by-point response, with related information on how it has been addressed in the revised version of the manuscript.

Reviewer #1 (Remarks to the Author):

Martínez-Mármol et al. report in their manuscript that Tau forms presynaptic nanoclusters and regulates the dynamic of the recycling synaptic vesicle pool. After identifying Tau as a protein that gets phosphorylated upon hyperexcitation and being a phase separating IDP in neurons, they reason that Tau might be a regulator of SV dynamics, in analogy to Synpsin1. Using single molecule particle tracking techniques, the authors compare the trajectories of Tau-mEOS2 in presynapses with that of axonal Tau.

They find that Tau forms 'nanoclusters' in presynapses and seems to regulate the decrease the diffusion dynamics of recycling synaptic vesicles. The authors then argue that the Tau clusters are biomolecular condensates based on their sensitivity to hexanediol.

This is a nice story showing the clustering of Tau in presynapses and the effect on vesicle dynamics, however, besides these two aspects most reported findings have previously been shown. The story lacks novelty and conclusive experiments that support the conclusion of biomolecular condensation of Tau in the presynapse.

Major concerns:

- The claimed formation of biomolecular condensates of Tau in presynapses - which by definition in the Tau and phase separation communities should have liquid-like properties - is not supported by the data, and the title and conclusion is therefore misleading. The HEK cell images showing large dots of GFP-Tau do not support the conclusion of condensation either bc they

look rather lysosomal. Those are likely not Tau condensates. Using sensitivity to hexandiol as the only sign of being a liquid condensates is not accepted because HD changes and dissolves a lot of things in cells!

We are sorry to hear that the reviewer does not think that the large Tau-GFP-positive dots that were detected in HEK-293T cells are condensates, despite the fact that (1) they were sensitive to 1,6-hexanediol, (2) their size and number increased with the amount of DNA transfected and (3) their recovery after photobleaching was much slower than in the cytosol.

To demonstrate the phase condensate nature of intracellular Tau-GFP droplets we have carried out several additional experiments:

- We performed additional immunocytochemistry in HEK-293T cells expressing Tau-GFP with a range of markers of membrane-bound organelles, such as endosomes, the endoplasmic reticulum, and lysosomes. We demonstrate that the high-intensity droplets do not co-localise with lysosomal markers or other endosomal markers as feared by the reviewer. The results of these experiments are included in Suppl. Fig. S8 f-h.
- We used two additional mutants that either partially (Tau^{I277P / I308P}) or fully (Δ Tau74) inhibit Tau's ability to generate condensates (Wegmann et al, 2018; Martinez-Marmol et al., 2023) to test this property in HEK-293T cells. We found that the number of Tau-GFP-positive droplets was significantly lower for both mutants compared to Tau^{WT}, and their size was also reduced, especially for the Δ Tau74 mutant. These results are in good agreement with their known ability to prevent the formation of biomolecular condensates. We have included this additional data in Suppl. Fig. S9 d-g.
- We have carried out additional FRAP experiments in Tau KO hippocampal neurons re-expressing Tau-GFP, to complement the pre-existing data set in HEK-293T cells. We found that the Tau-GFP droplets in neurons also displayed a much-reduced recovery after photobleaching compared to Tau-GFP in the cytosol. We have included the results of these experiments in Suppl. Fig. S10 a-e.
- In addition to the immunocytochemistry carried out in HEK-293T cells, we also performed additional immunocytochemistry experiments in Tau KO hippocampal neurons re-expressing Tau-GFP. We demonstrate that the high-intensity Tau droplets do

not co-localise with lysosomal markers or other endosomal markers in neurons. The results of these experiments are included in Suppl. Fig. S10 f-h.

- We carried out new sptPALM analyses of these Tau mutants in Tau KO hippocampal neurons and examined the impact of these mutations/deletion on Tau-mEos2 nanoclustering. We anticipated that if Tau formed nanoscopic presynaptic biomolecular condensates, preventing these from forming would largely increase Tau's mobility. We demonstrate that both Tau mutations (Tau^{I277P / I308P} and Δ Tau74) significantly increased Tau mobility and reduced its clustering. The results of these experiments are included in Fig. 5 g-k and Suppl. Fig. S12.

In conclusion, we have demonstrated that the identified large Tau-GFP droplets are biomolecular condensates that do not colocalise with lysosomes, which is in good agreement with previous reports (Wegmann et al., 2018). We further demonstrate that preventing Tau from forming biomolecular condensates largely increased its mobility and reduced its ability to form nanoclusters, indicating that Tau forms nanoscale biomolecular condensates at the presynapse and in axons. We thank the reviewer for their input and fair criticism, enabling us to generate additional supporting data and avoid sole reliance on 1,6-hexanediol.

- The introduction has too little information about Tau in general and presynaptic Tau, and important relevant landmark references are missing! e.g. general Tau reference and Tau on MT reference has to be given, reference(s) to Roland Brandt's lab who showed the binding activity of Tau to MTS by single particle tracking in 2014, and more recent work of his lab on Tau tracking in stress granules. These are important papers that report already many things presented as findings in this manuscript.

We have now included additional new sections in the introduction, providing references to the seminal work of Roland Brandt's lab.

- Activity dependent phosphorylation of Tau and its status of an IDP are known since a long time. The data presented here do not present any novel aspect of this, they do not show an increase in synaptic but in total Tau phosphorylation, and even lack the information about what kind of phosphorylation is found on Tau upon K⁺ induced neuronal activation. Neither is the increase of phosphorylation shown by IHC or Biochemistry.

We have now provided an analysis comparing the level of phosphorylation elicited by stimulation in synaptosomes and in hippocampal neurons. Combined with the known level of IDR per proteins identified, this allowed us to unveil candidates that could play a role in the generation of synaptic biomolecular condensates, showing Tau as one of the most disordered and highly phosphorylated proteins both in neurons and in synaptosomes. We have now completed additional phospho-proteomics analysis, including a comparison of all identified phosphosites between our neuronal phosphoproteome and the synaptosome phosphoproteome. We also provide a complete map of all Tau phosphosites identified, their conservation between our neuronal phosphoproteome and a synaptosome phosphoproteome dataset, and their phosphorylation level for both species and stimulation conditions. These additional analyses indicate that Tau is a novel candidate protein to be highly phosphorylated and undergo phase separation in neuronal synapses. The new data are presented in Fig. 2 Suppl. Figs S2 and S3 and Suppl. Table S1.

Reviewer #2 (Remarks to the Author):

Martinez-Marmol et al investigate the role of the Tau protein for synaptic vesicle organization and distribution in the presynaptic compartment. They make use of an exploratory phosphoproteomics approach, identifying Tau as a potential key player, and then use single-molecule super-resolution microscopy and biochemical assays to further characterize the role of Tau in activity-dependent regulation of synaptic vesicle nanoscale organization and function. They propose that liquid-liquid phase separation of Tau could play a central role in these processes.

The study is well-designed, the experiments are conducted thoroughly and the manuscript is well written. The authors address an important topic in neurobiology, how synaptic vesicles are organized at the presynapse. I thus recommend publication in Nature Communications after minor revisions, as outlined below:

- Expand or provide a more detailed analysis of the phosphoproteomics results: The phosphoproteomics data are presented in a protein-centric manner: A single phosphopeptide with the highest fold change per protein was used to show differential protein phosphorylation upon activity stimulation. However, proteins could also display a decrease in phosphorylation upon activity stimulation - this would be completely lost in this analysis. Also, multiple, reciprocal phosphorylation events on the same protein would be missed. For simplicity and to get an overview, the analysis of the authors is ok but we kindly ask the authors to include a complete overview of activity-dependent phosphorylation and dephosphorylation events on a phosphosite level in the Supplementary data (e.g. in a figure and a table). In addition, an overview of the phosphoproteomics data prior to filtering with SynGO in a supplementary figure would be appreciated.

We thank the reviewer for their kind comments on our manuscript. We have now performed all the requested analyses, presented in the new Fig. 2, Suppl. Figure S2, Suppl. Figure S3 and we provided all the data in Suppl. Table S1. We have now amended the result section with a much richer analysis, including (1) the comparison with the SynGO synaptic protein database, (2) a

complete overview of activity-dependent phosphorylation and dephosphorylation events on a phosphosite level, (3) a full comparison of all phosphosites between our neuronal phosphoproteome and the synaptosome phosphoproteome, and (4) a complete map of all Tau phosphosites identified, their conservation between our phosphoproteome and published rat synaptosome phosphoproteome data, as well as the phosphorylation status for both species and stimulation conditions. We thank the reviewer for requesting these analyses”. The amended section is copied as follows:

“Clustering of SVs has been hypothesized to depend on the formation of condensates through interactions between protein IDRs. Further, a multitude of synaptic proteins, particularly active zone scaffolding proteins, undergo a high level of activity-dependent phospho-regulation. Therefore, both activity-dependent phosphorylation and the level of structural disorder represent key metrics worth joint consideration when identifying candidate proteins capable of controlling the mobility of recycling vesicles via LLPS. Accordingly, we used mass spectrometry analysis to obtain the phosphoproteome of mouse hippocampal neurons stimulated with high K⁺, identifying 20,395 phosphopeptides, of which 4,626 differed significantly in their phosphorylation status (threshold adjusted $P < 0.05$) in the stimulated condition compared to the resting control condition (Supplementary Table S1). An enrichment dataset of the proteins containing 500 of the most significant differentially phosphorylated peptides from stimulated hippocampal neurons was compared with the SynGO synaptic protein database 45 (Fig. 2 a and Supplementary Fig. S2 a). SynGO analysis revealed the presence of pre- and postsynaptic proteins, including postsynaptic density constituents and the presynaptic active zone, among the phosphorylated proteins identified (Supplementary Fig. S2 a). It also highlighted a strong representation of proteins associated with mechanisms involved in the structural organization of the synapse and the recycling of SVs (Fig. 2 a), supporting an activity-dependent phospho-regulation of proteins involved in synapse organization. To further confirm a synaptic enrichment in our phosphoproteomic analysis, a total 4,626 phosphopeptides from the obtained mouse hippocampal phosphoproteome were compared with a phosphoproteome dataset containing 1,917 phosphopeptides from stimulated rat synaptosomes 44. The sequence of each homologous protein from mouse neurons and rat synaptosomes was fully aligned to determine phosphosite detection and conversion. This analysis indicated that synaptosome phosphosites responding to stimulation were also identified in the stimulated mouse total hippocampal neurons (Fig. 2 b). The smaller number of phosphosites identified in the rat synaptosomal preparation compared to that of the entire hippocampal mouse phosphoproteome may be due to

the synaptic enrichment of the subcellular rat synaptosomal phosphoproteome or to technical advances in phosphoproteome coverage. Given these results, we next devised an unbiased correlative approach to identify highly intrinsically disordered proteins that undergo activity-dependent phosphorylation. For each presynaptic protein, the phosphopeptide with the highest upregulated phosphorylation value was used to assign a maximum log₂ fold change (Fig. 2 c). Candidate proteins were then identified by comparing the percentage of disorder with the activity-dependent increase in phosphorylation level for the mouse phosphoproteome (Fig. 2 c, Supplementary Table S1). We refined our analysis by including the phosphoproteome data from stimulated rat synaptosomes (Supplementary Fig. S2 b, Supplementary Table S1). Among the identified candidate proteins, Tau, Bassoon, Synapsin 1, β -synuclein, Myristoylated Alanine Rich C-Kinase Substrate (MARCKS) and MARCKS-like protein (MLP) were some of the most highly disordered presynaptic proteins that also exhibited significant up-regulation of phosphorylation in response to stimulation. Tau primarily works as an axonal protein controlling the transport of vesicles. However, its function within synaptic terminals remains largely unexplored. Natively, Tau is an intrinsically disordered protein that, like Synapsin 1, has been shown to form biomolecular condensates in vitro and in situ 18 within neurons. Tau is phosphorylated in an activity-dependent manner, regulating its association with microtubules and the formation of biomolecular condensates. Furthermore, our analysis showed that >90% of all mouse Tau phosphosites were identified between mouse and rat synaptosomes, of which >50% demonstrated conserved phosphorylation directionality in response to stimulation, including many of the most differentially phosphorylated phosphopeptides (Fig. 2 b and Supplementary Fig. S2 c; a full map of each mouse Tau phosphosite and phosphorylation status in rat synaptosomes is provided in Supplementary Fig. S3). The mouse hippocampal phosphoproteome data exhibited the same phosphorylation site changes as presynaptic proteins known to undergo activity-dependent phosphorylation, such as the decreased phosphorylation of dynamin 1 at S774, and synapsin 1 at S62, as well as multiple increases and decreases of tau phosphorylation levels (Fig. 2 d and Supplementary Fig. S2 c). Tau provided 53 peptides whose phosphorylation is regulated (up or down) in response to different types of stimulation both in neurons and synaptosomes (Supplementary Fig. S3).”

- Show controls for or comment on Tau liquid-liquid phase separation experiments in HEK cells: The authors should comment on their experimental design when overexpressing Tau in HEK cells with a GFP-tag, and include a separate experiment with a similar-sized control protein at similar expression levels not showing phase separation in the same conditions (or provide a similar reference). Such a control would significantly strengthen the manuscript.

We agree that an independent protein would be great to use. However, for consistency, we took advantage of two Tau mutants characterized in Wegmann et al. 2018, that either partially (Tau^{I277P / I308P}) or fully (Δ Tau74) inhibit Tau's ability to generate condensates. These mutants were tested in HEK-293T cells and Tau KO neurons at the level of macroscopic Tau condensates generation, and nanoscopic Tau cluster formation. As anticipated, the number of Tau macroscopic droplets (condensates) was significantly lower for both mutants compared to Tau^{WT} and their size was also reduced, especially for the Δ Tau74 mutant. We have included this additional data in Suppl. Fig. S9 d-g. We also performed sptPALM analysis with these mutants in Tau KO hippocampal neurons and examined the impact of the mutation/deletion on Tau-mEos2 mobility and nanocluster formation. We anticipated that if Tau was forming nanoscale presynaptic biomolecular condensates, preventing these from forming would largely increase Tau's mobility. We demonstrate that the Tau mutants significantly increase Tau mobility and decrease the molecule's clustering, in good agreement with their known ability to prevent the formation of biomolecular condensate. The results of these experiments are included in Fig. 5 g-k, and Suppl. Fig. S12.

In conclusion, we have demonstrated that it is the ability of Tau to generate biomolecular condensates, rather than its expression level alone, that is needed to create the droplets. We further demonstrate that preventing Tau from forming biomolecular condensates largely decreases its clustering, indicating that Tau forms nanoscale biomolecular condensates at the presynapse and axons. We thank the reviewer for their input and fair criticism, which has enabled us to generate additional supporting data.

- Comment in more detail on the effects of phosphatase and kinase inhibitors in the Tau knockout: While the absence of OkAc-induced increase in recycling vesicle mobility in a Tau knockout is in line with the authors hypothesis, staurosporine treatment still lead to a decrease in recycling vesicle mobility. These observations could be discussed in the Discussion section, as they indicate additional components responsible for SV mobility which are dependent on phosphorylation but not on Tau.

In the same context, the authors could comment on the known targets of the kinase and phosphatase inhibitors used in this study.

We have now specifically discussed the use of these inhibitors in the discussion (p. 15, line 30), as requested.

- Provide more details for experimental conditions and data availability: In the sample preparation section, we found no mention of phosphatase inhibitors being used during sample preparation for the phosphoproteomics samples. Is this correct? Please also include the K concentrations of the "high/low conditions" earlier in the method section and mention them at least once in the results section (at present, the conditions are only mentioned once at the end of the method section).

Please also include access to the proteomics data uploaded to PRIDE, as required by the Nature Communications guidelines for authors.

We have now detailed the composition of the lysis buffer used for the proteomic preparation ("Protein digestion and phosphopeptide enrichment" p.40, line 7, of the Materials and Methods section). The lysis buffer does contain phosphatase inhibitors.

The potassium concentration of the high/low potassium buffers used in all the experiments can be found in the relevant section on super-resolution imaging ("Photoactivated localization microscopy (sptPALM), subdiffractional tracking of internalized molecules (sdTIM) and dual-color super-resolution imaging" p.36, line 1). The low K⁺ resting imaging buffer composition is 145 mM NaCl, 5.6 mM KCl, 2.2 mM CaCl₂, 0.5 mM MgCl₂, 5.6 mM D-glucose, 0.5 mM ascorbic acid, 0.1% BSA, 15 mM HEPES, pH 7.4). Neuronal stimulation was performed by incubating the neurons with high K⁺ stimulating imaging buffer (95 mM NaCl, 56 mM KCl, 2.2 mM CaCl₂, 0.5 mM MgCl₂, 5.6 mM D-glucose, 0.5 mM ascorbic acid, 0.1% BSA, 15 mM HEPES, pH 7.4).

The mass spectrometry proteomics data and MaxQuant output have been deposited to the ProteomeXchange Consortium via the PRIDE 92 partner repository with the dataset identifier PXD020232 and 10.6019/PXD020232 (p. 42, line 7).

Reviewer #3 (Remarks to the Author):

The work by Martinez-Marmol et al. uses relatively novel and highly sophisticated approaches for super-resolution single molecule tracking applied to different pools of synaptic vesicles (total using sptPALM or recycling sdTIM) as well as tracking tau in the axon and synapses of cultured neurons. These imaging approaches have facilitated some novel and impactful findings (noted below) that would advance the field. However, there are some issues that should be addressed. In particular, the entire focus on LLPS is not well supported and additional studies to further substantiate some aspects of the work and better discussion of the results is needed. Specific comments are provided below.

The work showing tau's role in regulating the mobility of the recycling pool of SVs is a novel finding and of high importance to better understanding the multi-functional roles of tau protein in neurons. Further, the differences in the properties of axonal tau clusters and synaptic tau clusters is an interesting novel finding. Surprising

The mass spectrometry work assessing the synaptic phosphoproteome with or without potassium stimulation is novel and provides compelling data on proteins modified in an activity-dependent fashion. However, there are no data convincingly demonstrating that the phosphorylation changes in tau are due specifically to synaptic tau as opposed to tau in any of the other cellular compartments (e.g. dendrites, cell bodies axons, etc.). The phosphorylation status of tau (e.g. sites and localization) was not assessed in the synaptic- and axon-focused experiments, which is a missed opportunity to provide potentially insightful data on specific tau phospho-changes associated with the different observations of tau behavior. Further, localization of pTau changes in the presynaptic space would strengthen this point and the involvement of synaptic tau phosphorylation in modulating SV mobility.

Generally, the data supporting the nanoclusters represent LLPS structures is not strong or well-supported in neurons, which is a central point of the manuscript. An area simply containing more protein than the surrounding area does not in and of itself constitute a LLPS structure. The nanocluster designation can stand alone without being described as LLPS structures, but if LLPS remains it must be further substantiated. The HEK cell work does not help to support neuronal LLPS and seems somewhat disconnected from the rest of the work presented in neurons.

Please see reply for reviewer #1 and #2. In brief, we have carried out the FRAP experiments in hippocampal neurons and found that the Tau-GFP droplets also displayed a much-reduced recovery after photobleaching compared to Tau-GFP in the cytosol, which is a hallmark of biomolecular condensates. We have included the result for this experiment in Suppl Fig. S10 a-e.

We took advantage of two Tau mutants characterized in Wegmann et al. 2018, that either partially (Tau^{I277P / I308P}) or fully (Δ Tau74) inhibit Tau's ability to generate condensates. These mutants were tested in HEK-293T cells and Tau KO neurons at the level of macroscopic Tau condensates generation, and nanoscopic Tau cluster formation. As anticipated, the number of Tau macroscopic droplets (condensates) was significantly lower for both mutants compared to Tau^{WT} and their size was also reduced, especially for the Δ Tau74 mutant. We have included this additional data in Suppl. Fig. S9 d-g.

We also performed sptPALM analysis with these mutants in Tau KO hippocampal neurons and examined the impact of the mutation/deletion on Tau-mEos2 mobility and nanocluster formation. We anticipated that if Tau was forming nanoscale presynaptic biomolecular condensates, preventing these from forming would largely increase Tau's mobility. We demonstrate that the Tau mutants significantly increase Tau mobility and decrease the molecule's clustering, in good agreement with their known ability to prevent the formation of biomolecular condensate. The results of these experiments are included in Fig. 5 g-k, and Suppl. Fig. S12.

In conclusion, in HEK-293T cells and neurons, we have demonstrated that Tau's ability to generate biomolecular condensates, rather than its expression level alone, is needed to create the observed droplets. We further demonstrate that preventing Tau from forming biomolecular condensates largely decreases its nanoclustering, indicating that Tau can form nanoscale biomolecular condensates at the presynapse and axons. We thank the reviewer for their input and fair criticism, which has enabled us to generate additional supporting data.

How were axons and synapses operationally defined? It is unclear how the authors confirmed (i.e. what metrics and/or markers were used) they were performing the tracking analyses in axons and bona fide synapses. Clarification and/or confirmatory studies are necessary.

Presynapses were identified using the presynaptic marker VAMP2-pHluorin. This allowed us to detect the unquenching of the pHluorin tag upon fusion of SVs, thereby identifying *bona fide* synaptic regions (Joensuu et al., JCB (2016); Joensuu et al., Nature Protocols (2017)). Perisynaptic regions adjacent to the fluorescent synaptic boutons were identified as axons/axonal segments.

There is no consideration/discussion of the focus on en passant synaptic structures versus terminal synaptic structures in the cultured neuron and whether differences may exist in tau's behavior/role in each. It is also not clear if the synapses being analyzed are functionally connected synapse and whether this could alter the observations made in these studies.

Hippocampal synapses have well-characterised "en passant" synapses, and we see no reason why Tau nanoclustering should differ in nerve terminals, except for the fact that they only have one route of entry into the presynapse, instead of a continuum of perisynaptic axon. This could potentially alter the kinetics of diffusion and ultimately the timing of Tau presynaptic clustering. We have included this point in the discussion (p. 16, line 12).

The other point about functional connections is fair, but unfortunately, our experiments do not allow us to distinguish between connected versus unconnected terminals. Based on high neuronal density and maturity, we assume neurons become highly connected as they mature and exhibit all the ultrastructural and electrophysiological hallmarks of connected synapses (Joensuu et al., JCB (2016); Joensuu et al., Nature Protocols (2017); Joensuu et al., EMBO J (2023)). However, unfortunately, we cannot exclude that among all the imaged nerve terminals, some could be non-functionally connected synapses, as our microscope is not fitted with an electrophysiology rig. Nevertheless, this should be highly mitigated by the elevated number of synapses analysed.

The introduction lacks introductory components on tau (a central component of the manuscript) and the discussion is extremely shallow.

We have now changed and extended both the introduction and the discussion.

There are a few minor misstatements: The epitope for Tau5 is incorrect (updated epitope - PMID: 17499212; original epitope - PMID: 8955115; original paper - PMID: 7479786). Tau is not exclusive to the nervous system.

We have now changed and corrected this sentence (p.38 line 19) and referenced the appropriate indicated papers.

REVIEWERS' COMMENTS

Reviewer #1 (Remarks to the Author):

The manuscript by Longfield et al describes how neuronal activity increases Tau and other synaptic protein phosphorylation and how it influences the mobility of Tau at microtubules and in presynapses, and how Tau influences the mobility of presynaptic vesicles. They reveal some very interesting new insights on which vesicle pools are affected by Tau, and identify that Tau forms nano-clusters both on microtubules and in presynaptic boutons that form only in conditions where previously reported biomolecular condensation of Tau has been observed.

The findings presented are novel and reveal first mechanistic details about the behavior of Tau in general and in its function of regulating presynaptic vesicle biology. They are therefore an important contribution to the understanding of Tau's role in neuronal function and allow new hypotheses on how Tau impairs synapses in neurodegenerative diseases.

Minor points that need to be addressed:

- 1) the assignment of phospho-sites in the Tau sequence seems to be done based on the longest human MAPT isoform, which peripheral "big Tau". In the brain, however, the longest human isoform (2N4R) has only 441 amino acids. The authors need to re-assign their p-sites according to the 2N4R Tau isoform in order to be able to identify which p-sites get phosphorylated and to be compatible with other research reports. All Figure panels related to these data need to be adjusted accordingly.
- 2) It would be very helpful for the reader if the authors could provide a small scheme about the structure of the presynapse and Tau in the beginning of the manuscript, and - more importantly - provide a model summarizing their findings in the end of the manuscript. This will make the large amount of data more understandable and interpretable.

Reviewer #2 (Remarks to the Author):

The minor points I had listed in the first round of reviews have been addressed adequately, both by generating more data as well as addition and modifications of text and figures. I thus recommend publication after addressing the very minor points below:

- In Figure 2D the right panel is too bulky. I'd suggest to reduce width of the p-site differential regulation list for clarity.
- the reviewer details for accessing the dataset were not added to the manuscript, only the PRIDE IDs which are yet private so cannot be accessed. I would kindly ask the authors to add the anonymous reviewer login details for private PRIDE datasets prior to publication that PRIDE generates automatically upon submission.

Point-by-point reply to the reviewers' comments and criticism (#NCOMMS – 22-26461B)

Reviewer #1 (Remarks to the Author):

The manuscript by Longfield et al describes how neuronal activity increases Tau and other synaptic protein phosphorylation and how it influences the mobility of Tau at microtubules and in presynapses, and how Tau influences the mobility of presynaptic vesicles. They reveal some very interesting new insights on which vesicle pools are affected by Tau, and identify that Tau forms nano-clusters both on microtubules and in presynaptic boutons that form only in conditions where previously reported biomolecular condensation of Tau has been observed.

The findings presented are novel and reveal first mechanistic details about the behavior of Tau in general and in its function of regulating presynaptic vesicle biology. They are therefore an important contribution to the understanding of Tau's role in neuronal function and allow new hypotheses on how Tau impairs synapses in neurodegenerative diseases.

Minor points that need to be addressed:

1) the assignment of phospho-sites in the Tau sequence seems to be done based on the longest human MAPT isoform, which peripheral "big Tau". In the brain, however, the longest human isoform (2N4R) has only 441 amino acids. The authors need to re-assign their p-sites according to the 2N4R Tau isoform in order to be able to identify which p-sites get phosphorylated and to be compatible with other research reports. All Figure panels related to these data need to be adjusted accordingly.

The reviewer is correct. The assignment of phospho-sites in the Tau sequences was performed based on the canonical mouse (*Mus musculus*) and rat (*Rattus norvegicus*) sequences, which correspond to the longest human Tau isoform. This isoform, called Big tau, is expressed mainly in the adult peripheral nervous system (PNS), but also in adult neurons of the central nervous system (CNS) that extend processes into the periphery (Fischer & Baas. *Trends Neurosci.* 2020). As requested by the reviewer and to facilitate the identification of the phospho-sites in the human most common Tau 2N4R isoform, we have now included the equivalent human Tau 2N4R residues in **Figure 2 c** and **Suppl. Fig. S3**. The incorporation of this additional information should help the identification of all detected phospho-sites with their homologous position in the 2N4R isoform, compatible with other research reports. We thank the reviewer for requesting this.

2) It would be very helpful for the reader if the authors could provide a small scheme about the structure of the presynapse and Tau in the beginning of the manuscript, and - more importantly - provide a model summarizing their findings in the end of the manuscript. This will make the large amount of data more understandable and interpretable.

We have now generated an additional figure (**Figure 7**) containing two schemes summarizing our findings to facilitate the interpretation of the results presented in the manuscript.

Reviewer #2 (Remarks to the Author):

The minor points I had listed in the first round of reviews have been addressed adequately, both by generating more data as well as addition and modifications of text and figures. I thus recommend publication after addressing the very minor points below:

- In Figure 2D the right panel is too bulky. I'd suggest to reduce width of the p-site differential regulation list for clarity.

We have now reduced the width of the p-site differential regulation and have filled some of the empty space with important information on the equivalent phospho-sites in the human Tau 2N4R isoform as requested by reviewer #1.

- the reviewer details for accessing the dataset were not added to the manuscript, only the PRIDE IDs which are yet private so cannot be accessed. I would kindly ask the authors to add the anonymous reviewer login details for private PRIDE datasets prior to publication that PRIDE generates automatically upon submission.

We have now made the PRIDE dataset available to the public. As this process usually takes a couple of days, if the reviewer wants to have full access now to this information, the account details are the following:

[REDACTED]